# Identification of nonsense-mediated decay inhibitors that alter the tumor immune landscape

Ashley L Cook[1,2], Surojit Sur[1,3,4], Laura Dobbyn[3], Evangeline Watson[1], Joshua D Cohen[1,3,4], Blair Ptak[3], Bum Seok Lee[3], Suman Paul[1,3,4], Emily Hsiue[1], Maria Popoli[1], Bert Vogelstein[1,2,3,4,5,6], Nickolas Papadopoulos[1,3,4], Chetan Bettegowda[1,2,3,4,7], Kathy Gabrielson[3,4,8], Shibin Zhou[1,3,4], Kenneth W Kinzler[1,2,3,4,6]*, Nicolas Wyhs[1,3,4]*

[1]Ludwig Center for Cancer Genetics and Therapeutics, Johns Hopkins University School of Medicine, Baltimore, United States; [2]Cellular and Molecular Medicine Graduate Program, Johns Hopkins University School of Medicine, Baltimore, United States; [3]Department of Oncology, Johns Hopkins Medical Institutions, Baltimore, United States; [4]Sidney Kimmel Cancer Center, Johns Hopkins University School of Medicine, Baltimore, United States; [5]Howard Hughes Medical Institute, Johns Hopkins University School of Medicine, Baltimore, United States; [6]Sol Goldman Pancreatic Cancer Research Center, Johns Hopkins University School of Medicine, Baltimore, United States; [7]Department of Neurosurgery, Johns Hopkins University School of Medicine, Baltimore, United States; [8]Department of Molecular and Comparative Pathobiology, Johns Hopkins University School of Medicine, Baltimore, United States

*For correspondence:
kinzlke@jhmi.edu (KWK);
nwyhs1@jh.edu (NW)

## eLife assessment

Here, the authors developed a cell-based screening assay for the identification of small molecule inhibitors of nonsense-mediated decay (NMD), and used it to validate KVS0001, a new small molecule SMG1 kinase inhibitor derived from the existing inhibitor SMG1i-11, showing it inhibits NMD in cultured cells leading to expression of neoantigens from NMD-targeted genes and slows tumor growth of cancer cell lines possessing a significant number of out-of-frame indel mutations. The conclusions are supported by **convincing** evidence, and the significance of this work consists in the development of a new and very promising NMD inhibitor drug that acts as an inhibitor of the SMG1 NMD kinase and is effective in animal tumor studies. This is an **important** advance for the field, as previous NMD inhibitors were not specific, lacked efficacy, or were very toxic and hence not suitable for animal applications.

**Abstract** Despite exciting developments in cancer immunotherapy, its broad application is limited by the paucity of targetable antigens on the tumor cell surface. As an intrinsic cellular pathway, nonsense-mediated decay (NMD) conceals neoantigens through the destruction of the RNA products from genes harboring truncating mutations. We developed and conducted a high-throughput screen, based on the ratiometric analysis of transcripts, to identify critical mediators of NMD in human cells. This screen implicated disruption of kinase SMG1's phosphorylation of UPF1 as a potential disruptor of NMD. This led us to design a novel SMG1 inhibitor, KVS0001, that elevates the expression of transcripts and proteins resulting from human and murine truncating mutations in vitro and murine cells in vivo. Most importantly, KVS0001 concomitantly increased the presentation

of immune-targetable human leukocyte antigens (HLA) class I-associated peptides from NMD-downregulated proteins on the surface of human cancer cells. KVS0001 provides new opportunities for studying NMD and the diseases in which NMD plays a role, including cancer and inherited diseases.

## Introduction

Despite success with cancer immunotherapies, approved immunotherapies are not available for the majority of cancer patients and only a minority of treated patients realize a durable response (*Haslam and Prasad, 2019*; *Ben-Aharon et al., 2018*; *Darvin et al., 2018*; *Yang et al., 2023*). Current studies largely focus on discovering new agents and identifying patients most likely to benefit from existing immunotherapies (*Kraehenbuehl et al., 2022*). While many studies correlate tumor insertion and deletion (indel) mutation load with immunotherapeutic response, not all tumors with high indel mutational loads respond to checkpoint inhibitors (*Mandal et al., 2019*; *Turajlic et al., 2017*; *Yarchoan et al., 2017*; *Chan et al., 2019*; *Van Allen et al., 2015*; *Samstein et al., 2019*; *Ma et al., 2022*; *Rizvi et al., 2015*).

The typical adult solid tumor contains a median of 54 coding somatic nucleotide variants, many of which have the potential to create novel neoantigens or Mutation-Associated NeoAntigens (*ICGC/TCGA Pan-Cancer Analysis of Whole Genomes Consortium, 2020*; *Vogelstein et al., 2013*; *Segal et al., 2008*). Approximately 5% of mutations are insertions/deletions (indels) or splice site changes that alter the open reading frame of the transcript (*ICGC/TCGA Pan-Cancer Analysis of Whole Genomes Consortium, 2020*; *Vogelstein et al., 2013*; *Segal et al., 2008*). This subset of mutations is of particular interest because they can result in proteins and derived peptides that are foreign to a host's healthy cells, giving rise to neoantigens (*Lindeboom et al., 2019*; *Pastor et al., 2010*; *Nogueira et al., 2021*). In normal cells, nonsense-mediated decay (NMD) plays an important role in messenger RNA (mRNA) quality control, as well as normal gene expression (*Frischmeyer and Dietz, 1999*; *Mendell et al., 2004*; *Carrard and Lejeune, 2023*; *Sun and Chen, 2023*). In cancer cells, however, NMD may aid immuno-evasion by eliminating RNA transcripts coming from genes that carry truncating mutants (*Nogueira et al., 2021*). This prevents translation and presentation of peptides from these proteins on major histocompatibility complex (MHC) class I complexes, rendering them invisible to immune cells (*Dalton et al., 2019*). Previous work has shown knockdown of the NMD pathway with small interfering RNA (siRNA) enhances the anti-tumor immune response (*Pastor et al., 2010*; *Castle et al., 2014*). Analysis of cell lines with loss of the UPF1 RNA helicase and ATPase gene (*UPF1*), a key mediator of NMD degradation, also showed increased levels of aberrant transcripts and mutant proteins in the alleles targeted by NMD (*Oka et al., 2021*). This is reminiscent of the pharmacological modulation of splicing, which has also been shown to increase the number of neoantigens present on the cancer cell surface due to similar underlying mechanisms (*Lu et al., 2021*). Despite early studies showing little toxicity with NMD inhibition, there is as of yet no reports of a specific chemical inhibitor of the pathway with good bioavailability (*Martin et al., 2014*; *Durand et al., 2007*; *Zhao et al., 2022*; *Gotham et al., 2016*; *Gopalsamy et al., 2012*).

In this work, we create a cell-based high-throughput assay to query the effects of a curated library of small molecules on NMD function. From this screen, we identify a lead compound capable of inhibiting NMD and ascertain its protein target as nonsense-mediated mRNA decay-associated phosphatidylinositol 3-kinase (PI3K)-related kinase (SMG1), a kinase that is a critical mediator of NMD. As the lead compound produced unacceptable toxicity in animal models, we then designed a specific and bioavailable small molecule inhibitor of SMG1 (KVS0001) that is well-tolerated in vivo. We then demonstrate that targeted inhibition of SMG1 by KVS0001 leads to the presentation of novel neoantigens identifiable by T-cells leading to tumor growth inhibition in vitro and in vivo.

## Results

### Development of a high-throughput assay to identify NMD inhibitors

To develop an assay to find NMD inhibitors, we identified isogenic cell lines with out-of-frame indel mutations, hereinafter referred to as truncating mutations, targeted by NMD activity (*Supplementary file 1*). We previously reported a panel of non-cancerous cell lines in which 19 common tumor

**eLife digest** Immunotherapies are treatments that have revolutionized cancer care by helping a patient's own immune system find and destroy cancer cells. Unfortunately, less than half of treated patients respond to these therapies, with tumors often learning to escape detection by the immune system.

One way that cancer cells can evade the immune system is by preventing themselves from producing mutant proteins. By stopping these proteins from reaching the cell surface, the abnormal cell is less likely to be detected and killed by the immune system. One way cancer cells accomplish this is by destroying the RNA templates needed to make the proteins through a process called 'nonsense-mediated decay'. Therefore, developing a therapy that can stop nonsense-mediated decay could help the immune system find and kill more tumor cells.

Cook et al. screened thousands of drugs with the aim of finding one that blocks nonsense-mediated decay. Although one drug was identified that could inhibit a gene called *SMG1* (which is known to activate nonsense-mediated decay), it was too toxic in animal models to be considered as a therapy. Therefore, Cook et al. developed a new drug targeting this gene that slowed tumor growth in mice without showing the same toxicity. Treating human cancer cells with the drug also increased the number of mutant proteins on the cell surface displayed to the immune system, suggesting the drug has the potential to prevent nonsense-mediated decay in humans.

The findings suggest that the drug developed by Cook et al. may make it easier for the immune system to identify and destroy certain cancer cells. This might also be relevant for other conditions involving nonsense-mediated decay, such as cystic fibrosis, Alport's disease, and Duchenne muscular dystrophy. If further studies confirm that the drug is safe and effective in humans, it could be used alongside cancer immunotherapies to improve patient response rates.

suppressor genes were inactivated using the Clustered Regularly Interspaced Short Palindromic Repeats (CRISPR)–Cas9 system (*Cook et al., 2022*). Through evaluation of this panel, we discovered two genes, Stromal Antigen 2 (*STAG2*) and Tumor Protein p53 (*TP53*), which did not express their expected proteins when assessed by western blots (STAG2) or immunohistochemistry (TP53) (*Figure 1—figure supplement 1*). Because the inactivation of these two genes was the result of frameshift mutations, we suspected that the absence of the proteins was due to NMD. This suspicion was supported by whole transcriptome RNA-sequencing (*Figure 1—figure supplement 2*; *Cook et al., 2022*). Notably, we saw an average decrease of *STAG2* RNA transcripts by 20-fold and *TP53* by 6-fold relative to their respective wild-type transcripts in the parental cell lines.

We selected two *STAG2* knockout clones (clones 2 and 8) and one *TP53* knockout clone (clone 221) derived from the Retinal Pigmented Epithelial (RPE1) cell line to design a high-throughput screen (HTS). Treatment with the canonical protein synthesis inhibitor emetine, working indirectly to inhibit NMD, demonstrated up to a 60-fold recovery of mutant RNA transcript expression, establishing this panel as appropriate for HTS (*Figure 1—figure supplement 3*; *Carter et al., 1995*).

We then designed a next-generation sequencing (NGS) assay for NMD (*Figure 1A*). We mixed the three cell lines in equal proportions and plated the mixture in 96-well plates followed by treatment with one compound (drug) per well. We determined NMD inhibition efficacy by comparing the wild-type and truncating mutant transcript expression levels in a ratiometric manner. Specifically, the wild-type sequences of the reciprocally knocked out clone (i.e., wild-type *STAG2* sequence from the *TP53* knockout clones, and wild-type *TP53* sequence from the *STAG2* knockout clones) served as internal references, providing a ratiometric assay of mutant to wild-type transcript abundance. This ratiometric assay minimized confounders introduced by nonspecific transcriptional activators or generally toxic agents. The use of three cell lines with different truncating mutations from two different target genes minimized the possibility that drugs identified in the HTS were cell line clone or mutation specific. Note that the use of these cell lines, carefully mixed, banked, and preserved, did not substantially increase the amount of time or work required to screen a single-cell line. A combination of well and plate barcodes allowed the pooling and scoring of over 1920 assays in a single NGS lane (see methods).

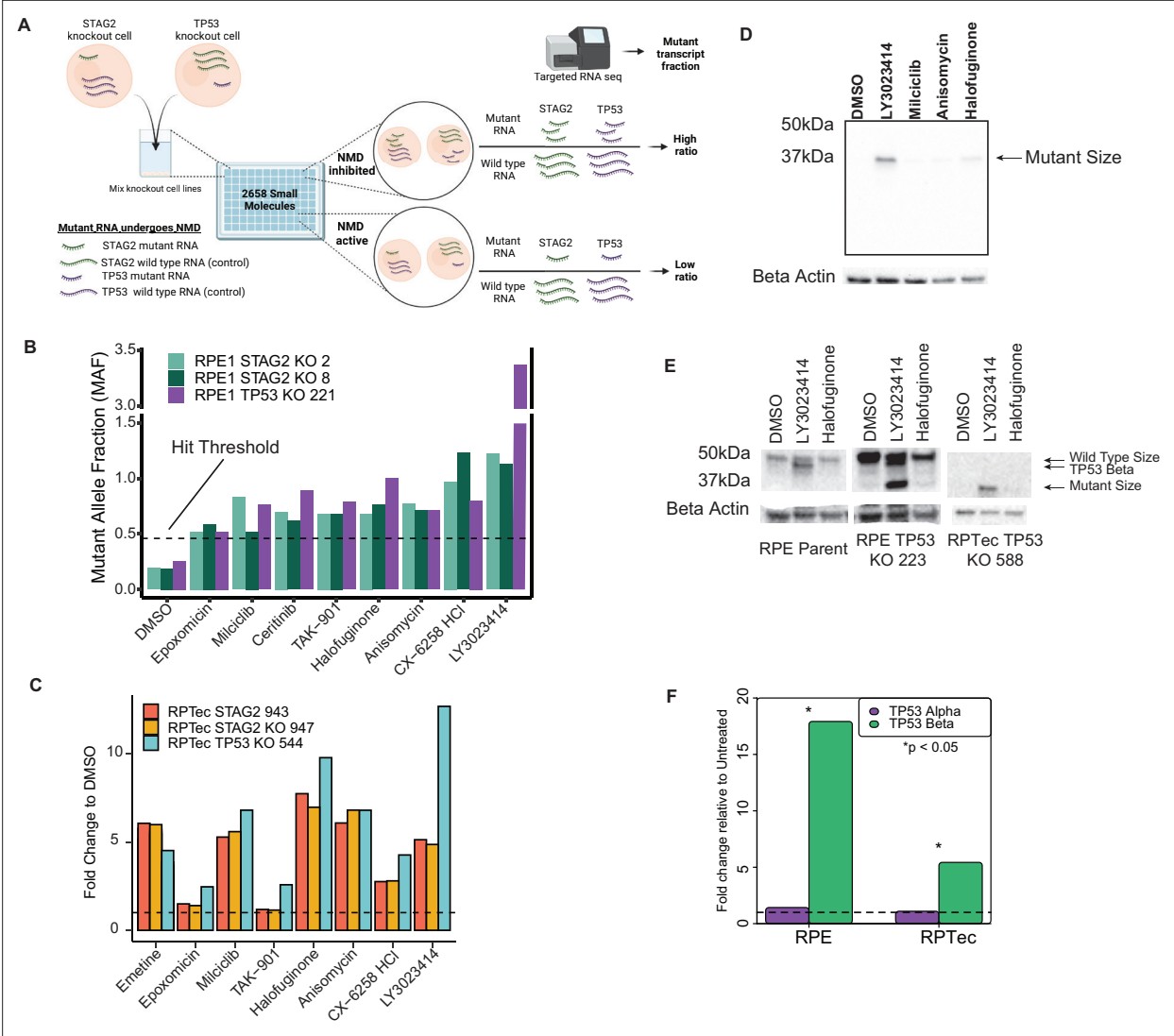

**Figure 1.** LY3023414 is a small molecule capable of increasing transcription of nonsense-mediated decay (NMD) targets. (**A**) Schematic of high-throughput screen (HTS) used to identify inhibitors of NMD. Mutant transcripts are represented by a smaller length in the cartoon for illustrative purposes only. All small molecules were tested at 10 µM. (**B**) Mutant RNA reads relative to wild-type reads for the top 8 hits from the HTS. The dotted line represents the minimum fraction required to be considered a hit (>5 standard deviations above dimethyl sulfoxide [DMSO] control). Full screen results are presented in *Figure 1—figure supplement 4*. (**C**) Targeted RNA-sequencing results of isogenic RPTec knockout clones treated with the eight best hits from the HTS at 10 µM. The dotted line represents a relative RNA expression level of 1, equal to that of DMSO-treated wells. Data for ceritinib, which did not validate on any line, are presented only in *Figure 1—figure supplement 8*. (**D**) TP53 western blot on RPE TP53 224, containing a homozygous *TP53* mutation, using the four hit compounds that validated in RPTec isogenic lines at 10 µM. (**E**) Western blot analysis of full-length TP53α and isoform TP53β after treatment with two NMD inhibitor lead candidates at 10 µM. TP53β (expression known to be controlled by NMD) as well as mutant TP53 are prominently induced by LY3023414 whereas full length is not. Note that RPE TP53 223 is a heterozygous knockout clone with one near wild-type allele whereas RPTec TP53 588 contains a homozygous *TP53* indel mutation. (**F**) Quantitative real-time PCR (qPCR) showing 10 µM LY3023414 treatment causes increased expression of the NMD controlled alternative transcript for *TP53*, *TP53β*, in parent cell lines for RPE1 and RPTec. Significance determined by Student's *t*-test. Unless indicated otherwise cells were exposed to test compound for 16 hr.

The online version of this article includes the following figure supplement(s) for figure 1:

**Figure supplement 1.** Next-generation sequencing results depicting genomic mutations at the CRISPR target area in STAG2 (top left) and TP53 (top right) isogenic cell line clones in the RPE1 cell line which were used in subsequent experiments.

**Figure supplement 2.** Fold change in RNA expression levels from whole transcriptome RNA-sequencing data for STAG2 and TP53 knockout clones in the RPE1 cell line background.

**Figure supplement 3.** RNA transcript level changes based on quantitative real-time PCR (qPCR) in STAG2 and TP53 knockout clones treated with the known nonsense-mediated decay (NMD) inhibitor emetine at 12 mg/ml.

*Figure 1 continued on next page*

*Figure 1 continued*

**Figure supplement 4.** Primary screen results from high-throughput assay.

**Figure supplement 5.** Emetine and dimethyl sulfoxide (DMSO) control sample data from the high-throughput screen (HTS) for each of the three clones used as measured by deep-targeted RNA-sequencing.

**Figure supplement 6.** Knockout status for isogenic cell lines used in this study.

**Figure supplement 7.** Protein schematic cartoons showing indel mutation site and expected size of various TP53 knockout clones used in this study.

**Figure supplement 8.** Fold change in mutant RNA transcription levels for STAG2 and TP53 in three knockout cell lines from the RPtec background containing out-of-frame indels targeted by nonsense-mediated decay (NMD).

## Execution of an HTS to identify NMD inhibitors

Previous human clinical trials suggest that off-target toxicity at doses required for NMD inhibition makes emetine and other well-known NMD inhibitors unsuitable for human use (*Siddiqui et al., 1973*; *Moertel et al., 1974*; *Mastrangelo et al., 1973*; *Tang et al., 2012*; *Bongiorno et al., 2021*). We performed an HTS to identify more specific NMD inhibitors by treating the isogenic cell line panel described above with a commercially available library consisting of 2658 FDA-approved or in late-phase clinical trial small molecules and natural products (*Supplementary file 2*). After purifying RNA from cells 16 hr post-treatment, we scored NMD inhibition using the strategy described in *Figure 1A* and *Figure 1—figure supplement 4*. Predictably, emetine increased the relative expression of mutant to wild-type transcripts by three- to fourfold on average (*Figure 1—figure supplement 5*). Eight compounds (0.3% of the library) increased the ratiometric mutant transcript fraction more than 5 standard deviations above the dimethyl sulfoxide (DMSO) controls in all three cell lines (*Figure 1B*). This hit threshold was chosen as it was the minimum required to ensure no false positives were observed in the DMSO controls. One of these eight compounds, anisomycin, is a known inhibitor of protein synthesis and commonly used NMD inhibitor for in vitro studies, providing independent validation of the screen (*Carter et al., 1995*). The other seven compounds increased mutant RNA transcript levels five- to tenfold relative to untreated cells.

## LY3023414 inhibits NMD and causes re-expression of mutant RNA and protein

To validate the eight hit compounds described above, we tested their effects in additional lines with mutations targeted by NMD (*Supplementary file 1*). First, we assessed them on isogenic *STAG2* and *TP53* knockouts in RPtec cells, another non-cancerous cell line (*Figure 1—figure supplement 6A, B* and *Figure 1—figure supplement 7*; *Cook et al., 2022*). Four of the original eight hit compounds increased mutant RNA expression in a dose-dependent manner (*Figure 1C* and *Figure 1—figure supplement 8*). Next, we examined the effects of these four compounds on additional RPE1 TP53 knockout cell lines with different mutations predicted to generate truncated TP53 proteins (*Figure 1—figure supplement 6C* and *Figure 1—figure supplement 7*). While treatment with all four potential NMD inhibitors restored expression of the truncated mutant TP53 proteins, two of them (LY3023414 and halofuginone) did so most robustly (*Figure 1D, E*). Additionally, *TP53* has an isoform, *TP53β*, whose expression is known to be controlled by the NMD pathway (*Cowen and Tang, 2017*). Using cell lines with intact *TP53β* isoform transcripts, we observed an increase in the *TP53β* isoform in cell lines treated with both LY3023414 and halofuginone (*Figure 1E*, middle and left columns). Quantitative real-time PCR (qPCR) of both the full-length *TP53*- (*TP53α*) and NMD-sensitive (*TP53β*) transcripts in parental RPE1 and RPTec cells treated with LY3023414 or halofuginone confirmed the selective upregulation of *TP53β*, but not *TP53α*, transcripts (*Figure 1F*). Based on a consistently stronger effect of LY3023414 over halofuginone across multiple isogenic cell lines and assays, LY3023414 was chosen to be the initial lead compound for further studies involving NMD inhibition.

## LY3023414 increases expression of NMD repressed mutant RNA transcripts and proteins in vitro and in vivo

To evaluate whether LY3023414 could relieve NMD repression of naturally occurring heterozygous mutant transcripts, we chose the NCI-H358 and LS180 cancer cell lines (*Supplementary file 1*). Treatment with LY3023414 followed by whole transcriptome RNA-sequencing revealed increased expression of the mutant allele in 42% and 67% of heterozygous, out-of-frame, indel mutations in these two

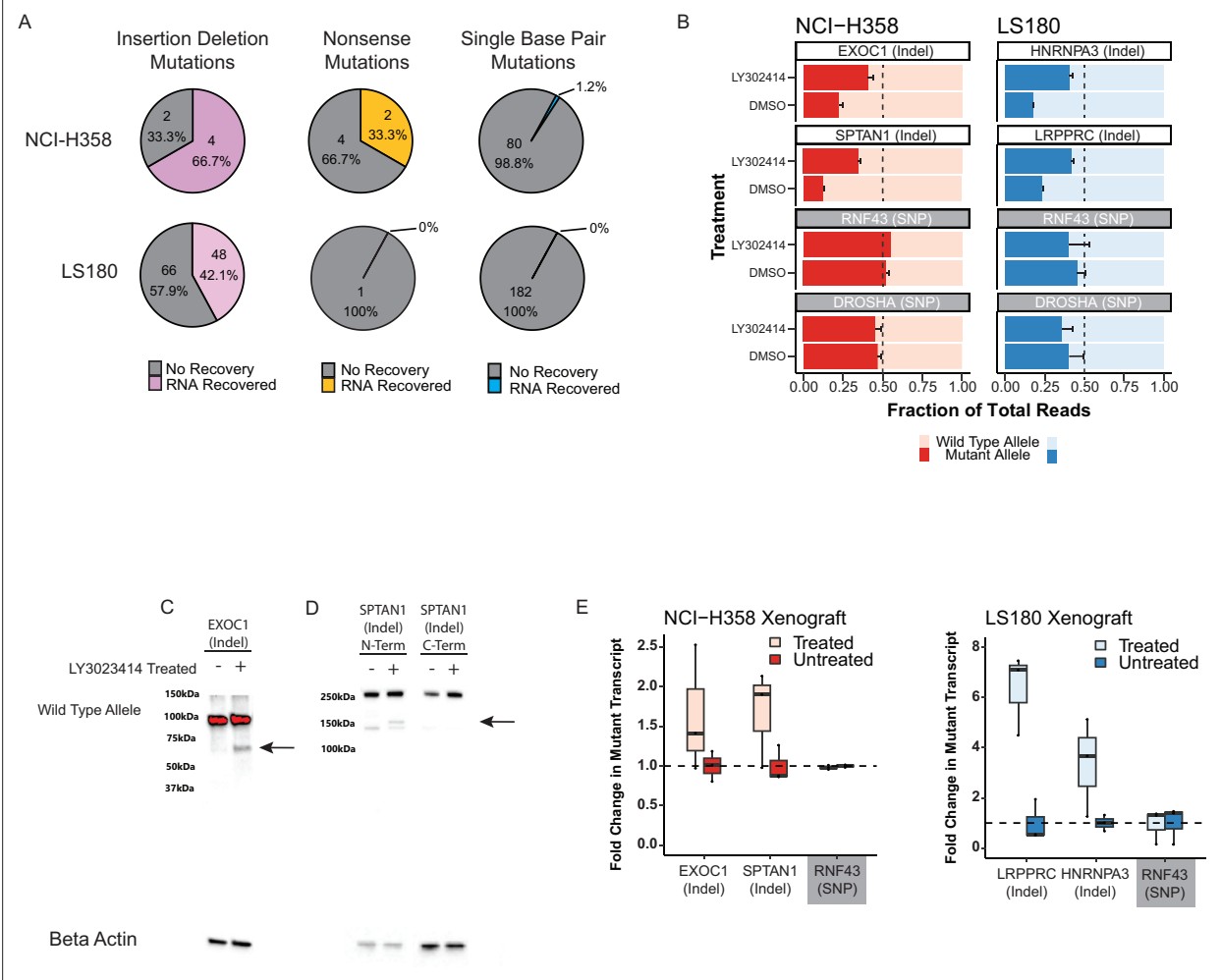

**Figure 2.** Inhibiting nonsense-mediated decay (NMD) in cancer cells increases broad expression of truncated gene messenger RNA (mRNA) and protein. (**A**) Mutant transcript recovery rates for genes containing heterozygous indel mutations based on RNA-sequencing results in cell lines treated with 5 µM LY3023414 for 16 hr. Strict inclusion criteria were used, such that only mutations with sufficient sequencing coverage are shown (see methods). Recovery is defined as at least two-fold increase over dimethyl sulfoxide (DMSO) treatment. (**B**) Targeted high coverage RNA-sequencing confirms recovery of mutant transcript levels in NCI-H358 and LS180 cancer cell lines treated with 5 µM LY3023414. RNF43 and DROSHA contain common heterozygous single-nucleotide polymorphisms (SNPs) and the mutant allele refers to the non-reference genome allele. Error bars indicate 95% confidence limits. (**C**) Western blot analyses of NCI-H358 cells showing mutant and wild-type protein levels in EXOC1 and (**D**) SPTAN1 with and without 5 µM LY3023414 treatment. The black arrow indicates the expected size of the mutant protein. The C-terminal SPTAN1 antibody is downstream of the out-of-frame indel mutation and is not expected to identify the mutant allele. (**E**) Fold change in the number of mutant RNA transcripts from deep-targeted RNA-sequencing of heterozygous mutated genes in NCI-H358 and LS180 xenografts treated by oral gavage with 60 mg/kg LY3023414 assayed 16 hr post-treatment. Student's *t*-test for target genes are all p < 0.05, while the null hypothesis holds for RNF43 (common SNP).

lines (*Figure 2A*). A third of the nonsense mutations in NCI-H358 were also 'recovered' (i.e., mutant transcripts increased relative to wild-type transcripts) after treatment with LY3023414 (*Figure 2A*, middle). In LS180, there was only one nonsense mutation meeting the required coverage, so it could not be evaluated in depth. Single base pair substitutions not resulting in stop codons were not affected by LY3023414 in either line (*Figure 2A*, right).

The whole transcriptome sequencing data were confirmed by targeted deep RNA-sequencing of two naturally occurring heterozygous truncating mutations from each cell line. Increased expression of the truncated mutant alleles relative to the wild-type alleles of the *EXOC1* and *SPTAN1* genes was observed in NCI-H358 cells after treatment with LY3023414 (*Figure 2B*, left). Likewise, increased expression of *HNRNPA3* and *LRPPRC* mutant transcripts was observed in LS180 after LY3023414 treatment (*Figure 2B*, right). We noted no significant effects on two coding region heterozygous single-nucleotide polymorphisms (SNPs) after LY3023414 treatment in the genes *RNF43* or *DROSHA*

in either cell line (*Figure 2B*). To substantiate these transcriptomic effects at the protein level, we treated NCI-H358 cells for 24 hr with LY3023414. Western blotting revealed truncated mutant protein for both EXOC1 and SPTAN1 after treatment with LY3023414, whereas the full-length protein levels remained present and unchanged in both the treated and control (DMSO) samples (*Figure 2C, D*, left) (*Williams et al., 2010*). An antibody that recognizes the C-terminus of SPTAN1, encoded downstream of the frameshift mutation, did not detect mutant protein, as expected (*Figure 2D*, right).

LY3023414 was originally developed as a PI3K inhibitor with activity against AKT Serine/Threonine Kinase 1 (AKT1) and Mammalian Target of Rapamycin (mTOR) (*Smith et al., 2016*). It was tested in a number of clinical trials and has a well-known pharmacokinetic profile in vivo (*Bendell et al., 2018*; *Rubinstein et al., 2020*; *Wei et al., 2016*; *Zou et al., 2017*; *Sweeney et al., 2022*). To determine whether LY3023414 affects NMD in vivo, we established xenograft tumors of both NCI-H358 and LS180 in nude mice. Treatment of these mice with a single oral dose of 60 mg/kg LY3023414 led to a significant increase in the expression of mutant RNA transcripts relative to wild-type transcripts 16 hr later (*Figure 2E*). The *RNF43* gene, which harbors a coding region heterozygous SNP, served as a control (*Figure 2E*). Severe drug-associated toxicity, including weight loss bordering on cachexia and near total inactivity, precluded the longer-term dosing required for anti-tumorigenic effects of LY3023414 in both BALB/c and C57BL/6N immunocompetent mouse strains.

## The kinase SMG1 is the target for NMD inhibition by LY3023414

To investigate the mechanism of NMD inhibition by LY3023414, we evaluated the six kinases with the highest reported inhibition by LY3023414 (*Smith et al., 2016*). siRNA-mediated knockdown of each of these kinases in NCI-H358 and LS180 cancer cell lines was performed for this purpose (*Figure 3—figure supplement 1*). Only knockdown of *SMG1* resulted in significant changes in the amount of truncating mutant transcript relative to wild-type transcript in all four genes evaluated (*Figure 3A*, top). *RNF43* and *DROSHA* contain heterozygous coding region SNPs and served as controls in these experiments and as expected showed no changes despite siRNA treatment (*Figure 3A*, bottom).

*SMG1* is known to regulate the NMD pathway by activating *UPF1*, an enzyme with RNA helicase and ATPase activity (*Yamashita et al., 2001*; *Yamashita, 2013*). We therefore knocked down *UPF1* with siRNA and found that it restored expression of the NMD-downregulated transcripts, at levels similar to those observed after the knockdown of *SMG1* (*Figure 3B*). Additionally, treatment of NCI-H358 and LS180 cancer cell lines with a previously reported SMG1-specific small molecule inhibitor, SMG1i-11, resulted in specific increases in the transcripts and proteins from genes with truncating mutations, just as did LY3023414 (*Figure 3—figure supplements 2 and 3*; *Gopalsamy et al., 2012*). Although SMG1i-11 displayed considerable SMG1 specificity and marked inhibition of NMD, it was highly insoluble. While we had no difficulty getting it into solution in DMSO for in vitro work, we were unable to find a vehicle to administer it in vivo, despite numerous attempts at various formulations and administration routes. This may explain why SMG1i-11 has been demonstrated to be an effective SMG1 inhibitor in vitro, but no peer-reviewed reports of its in vivo activity have been reported to date (*Gopalsamy et al., 2012*; *Keenan et al., 2019*; *Valley et al., 2019*).

## Development of an improved NMD inhibitor targeting SMG1

Although we were unable to secure a viable lead compound, the HTS did identify *SMG1* as an ideal target to disrupt the NMD pathway. We sought to develop a new SMG1 inhibitor based on the cryo-electron microscopy structure of the binding pocket of SMG1 (*Zhu et al., 2019*; *Langer et al., 2021*). We attempted the synthesis of eleven compounds (KVS0001 to KVS0011) and tested for bioavailability and preservation of target specificity. Among these, KVS0001 stood out due to its solubility while preserving SMG1 inhibitory activity (*Supplementary file 3*, *Figure 3C*, and *Figure 3—figure supplement 4*). Mass spectrometry-based assays showed that KVS0001 inhibits SMG1 protein more than any of the other 246 protein or lipid kinases tested at concentrations from 10 nM to 1 µM (*Supplementary file 4* and *Figure 3—figure supplement 5*; *Patricelli et al., 2011*). Noteworthy off-target kinase inhibition was not observed until doses of 1 µM and above. Experiments using NCI-H358 and LS180 cells showed that KVS0001 is bioactive in the nanomolar range and subverts the NMD-mediated downregulation of truncating mutant transcripts and proteins (*Figure 3D, E*, *Figure 3—figure supplements 6 and 7*). Inhibition at concentrations as low as 600 nM led to equal expression of wild-type and mutant transcripts, suggesting near total blockade of the NMD pathway (*Figure 3D*,

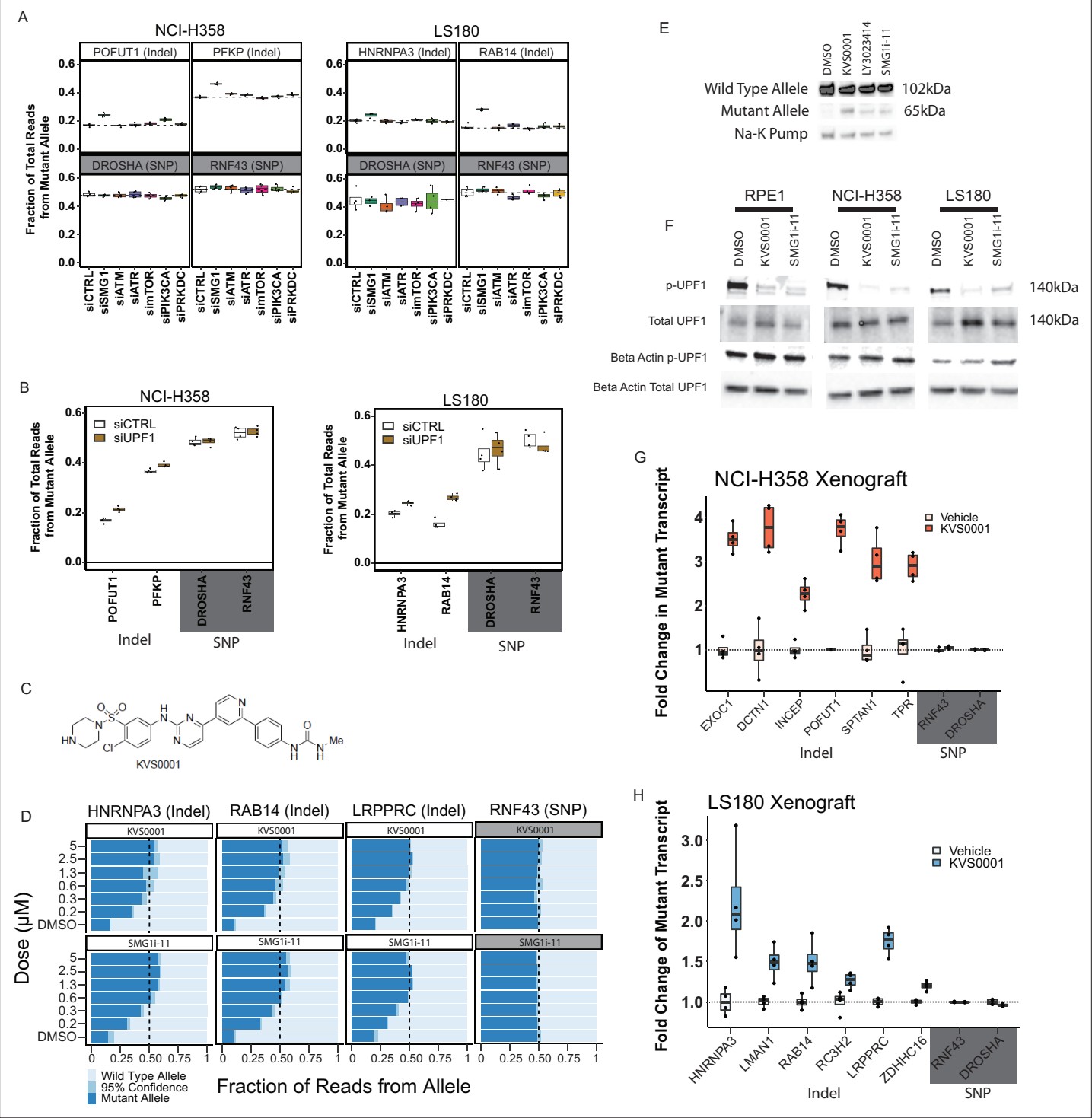

**Figure 3.** Novel nonsense-mediated decay (NMD) inhibitor KVS0001 is SMG1 specific and induces expression of NMD-targeted genes in vitro and in vivo. (**A**) Fraction of mutant allele transcripts in genes with heterozygous indels previously established in this study as sensitive to NMD inhibition. Results show mutant levels after siRNA treatment targeting kinases inhibited by LY3023414. RNF43 and DROSHA are common heterozygous single-nucleotide polymorphisms (SNPs) (shaded gray) and serve as negative controls. (**B**) Fraction of mutant allele transcripts in genes with truncating mutations known to be sensitive to NMD inhibition after siRNA treatment with siUPF1 or non-targeting siRNA. Data from deep-targeted RNA-sequencing. (**C**) Structure of novel NMD inhibitor KVS0001. (**D**) Targeted RNA-sequencing on three genes with heterozygous, out-of-frame, indel mutations in LS180 cancer cells treated in a dose–response with KVS0001 or SMG1i-11. RNF43 serves as a control (common heterozygous SNP) and the mutant allele refers to the non-reference genome allele. (**E**) Western blot of EXOC1 protein in NCI-H358 cells treated with 5 μM novel inhibitor

*Figure 3 continued on next page*

*Figure 3 continued*

KVS0001, LY3023414, or SMG1i-11 for 24 hr. (**F**) Western blot of phosphorylated UPF1 on three cell lines treated with 5 μM KVS0001, SMG1i-11, or dimethyl sulfoxide (DMSO). Note that total UPF1 and p-UPF1 were run on different gels, loading controls correspond to indicated gel. (**G**) Fold change in the number of mutant allele transcripts measured by targeted RNA-seq in genes containing heterozygous out-of-frame indel mutations in NCI-H358 or (**H**) LS180 subcutaneous xenografts in bilateral flanks of nude mice. Mice were treated once with intraperitoneal (IP) injection of vehicle or 30 mg/kg KVS0001 and tumors harvested 16 hr post IP treatment. All genes shown contain heterozygous out-of-frame truncating mutations except RNF43 and DROSHA which serve as controls (contain heterozygous SNPs).

The online version of this article includes the following figure supplement(s) for figure 3:

**Figure supplement 1.** RNA expression levels of kinases post siRNA targeting in NCI-H358 and LS180 cells by quantitative real-time PCR (qPCR).

**Figure supplement 2.** Slider plots showing mutant allele fraction relative to total reads measured by deep-targeted RNA-sequencing for in vitro treated LS180 (top in red) or NCI-H358 (bottom in blue) cells with 5 μM LY3023414 (labeled LYO) or a previously described SMG1 inhibitor SMG1i-11 at 1 μM.

**Figure supplement 3.** Mutant protein expression only occurs in the presence of nonsense-mediated decay (NMD) inhibtion.

**Figure supplement 4.** Biophysical properties for novel SMG1 inhibitor KVS0001.

**Figure supplement 5.** Kinase specifity for KVS0001.

**Figure supplement 6.** (**A**) Slider plots showing mutant allele fraction measured by deep-targeted RNA-sequencing for genes from NCI-H358 (top in red) and (**B**) LS180 (bottom in blue) treated in a dose–response in vitro with novel nonsense-mediated decay (NMD) inhibitor KVS0001.

**Figure supplement 7.** Mutant protein expression in presence of nonsense-mediated decay (NMD) inhibtion.

---

*Figure 3—figure supplement 6*). Western blotting showed that KVS0001 substantially decreases the amount of phosphorylated UPF1, the downstream target of SMG1 kinase activity, in three different cell lines (*Figure 3F*). Finally, KVS0001 treatment of NCI-H358- and LS180-derived xenograft tumors in nude mice resulted in significant increases in transcript levels in each of six tested endogenous genes with truncating mutations, while having had no measurable effects on genes containing heterozygous coding region SNPs (*Figure 3G, H*).

## NMD inhibition with KVS0001 causes MHC class I display of hidden neoantigens

Cancer cells may evade immune surveillance by downregulating genes with truncating mutations as a result of NMD (*Nogueira et al., 2021*; *Hu et al., 2017*). Indeed, previous studies with non-specific or toxic NMD inhibitors have shown an increase in selected antigens from tumor-specific mutations when NMD is inhibited (*Becker et al., 2021*). We therefore investigated whether KVS0001 could similarly alter the cell surface presentation of proteins from genes harboring truncating mutations in NCI-H358 and LS180 cells. Using quantitative high-performance liquid chromatography (HPLC)–mass spectrometry (MS) we observed a striking (45- to 90-fold) increase in the EXOC1- and RAB14-derived neoantigens, and a significant (twofold) increase in the ZDHHC16-derived neoantigen (*Figure 4A, B*, *Figure 4—figure supplements 1 and 2*; *Wang et al., 2019*). This is consistent with the re-expression of mutant transcript and protein shown previously in this study (*Figure 3D, E*). These three peptides were chosen based on an in silico review of potentially presented peptides and a preliminary experiment that looked at qualitative (present or not present) presentation of the predicted peptides (*Reynisson et al., 2020*; *Schmidt et al., 2021*).

## Targetable peptide presentation occurs with NMD inhibition by KVS0001

To test whether cancer cell neoantigens presented as a result of NMD inhibition could be targeted, we evaluated two cancer cell lines, NCI-H716 and NCI-H2228 (*Supplementary file 1*). Both lines contain homozygous mutations in *TP53* which produce transcripts downregulated by NMD, with each exhibiting RNA levels less than 20% of the median level of expression among 675 cancer cell lines (*Figure 4C* and *Figure 4—figure supplement 3*; *Lindeboom et al., 2016*; *Klijn et al., 2015*). Treatment with KVS0001 increased the expression of TP53 protein in both lines, while the commonly used therapeutic agents 5-fluorouracil and etoposide did not affect TP53 abundance (*Figure 4D* and *Figure 4—figure supplement 4*).

To determine whether this disruption of NMD repression is targetable by T-cells, we developed a bispecific antibody (KVS-BI043) that recognizes a peptide–HLA complex on one end and CD3 on the other end. CD3 is expressed only on T-cells, and this bispecific antibody functions as a T-cell engager,

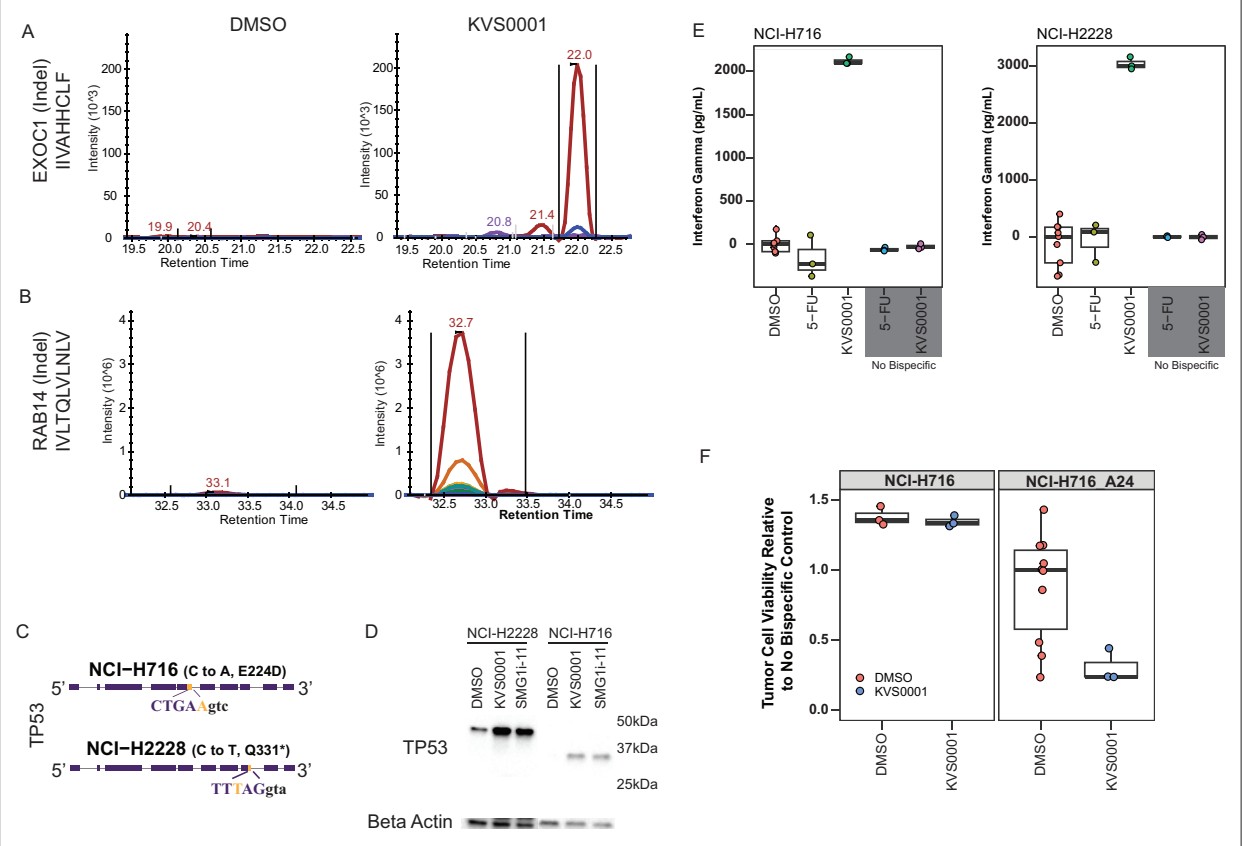

**Figure 4.** KVS0001 treatment induces targetable cell surface presentation of peptides known to be downregulated by nonsense-mediated decay (NMD). (**A**) MHC class I HLA presentation of mutant specific peptide sequences from NCI-H358 and (**B**) LS180 cells by quantitative HPLC–mass spectrometry treated with dimethyl sulfoxide (DMSO) or 5 µM KVS0001. The gene name, type of mutation (in parenthesis), and presented peptide are shown on the y-axis for each gene. Colors indicate different ions. (**C**) *TP53* gene structure and mutant DNA sequence for NCI-H716 and NCI-H2228 cancer cell lines, both contain a homozygous splice site mutation in *TP53*. Capital letters represent exonic sequence; lowercase letters represent intronic sequence. DNA mutation reflected by gold bases. (**D**) Western blot against TP53 in the presence or absence of 5 µM NMD inhibitor in NCI-H716_A24 and NCI-H2228 cell lines. NCI-H2228 has an expected size of 46.6 kDa and NCI-H716 of 34.7 kDa. (**E**) Interferon (IFN-γ) levels over baseline based on enzyme-linked immunosorbent assay (ELISA) in a co-culture assay with NCI-H716_A24 and NCI-H2228 cells, 1.25 µM NMD inhibitor, human CD3+ T-cells, and bispecific antibody for TP53 and CD3. Chemotherapy (5-fluorouracil) is shown as a control. (**F**) Cell killing based on luciferase levels in a co-culture assay in NCI-H716 cells with and without A24 expression, treated with TP53-CD3 bispecific antibody, 1.25 µM NMD inhibitor and human CD3+ T-cells.

The online version of this article includes the following figure supplement(s) for figure 4:

**Figure supplement 1.** DNA and protein sequences for the wild-type and mutant alleles of (**A**) EXOC1, (**B**) RAB14, and (**C**) ZDHHC16 genes.

**Figure supplement 2.** Heavy peptide loading controls and endogenous (light) peptide presentation of genes in LS180 and NCI-H358 treated with dimethyl sulfoxide (DMSO) or 5 µM KVS0001.

**Figure supplement 3.** Waterfall plot of publicly available TP53 RNA expression (as shown by FPKM) for 675 cancer cell lines.

**Figure supplement 4.** Western blot of TP53 on NCI-H716 and NCI-H2228 cells treated with 5 or 7.5 µM of nonsense-mediated decay (NMD) inhibitor, 1 µM SMG1i-11, or 200 mg/ml chemotherapy, showing controls related to *Figure 4D*.

linking target cells to cytotoxic T-cells, which then kill the targets (*Mack et al., 1995*; *Middelburg et al., 2021*). KVS-BI043 recognizes a ten amino acid peptide (residues 125–134 of TP53) bound to HLA-A24. NCI-H2228 naturally express A24 whereas NCI-H716 cells were engineered to express A24 using a retrovirus (NCI-H716_A24). Treatment of NCI-H716_A24 or NCI-H2228 cells with KVS0001 and the KVS-BI043 bispecific antibody in the presence of normal T-cells caused a significant increase in interferon (IFN-γ) release (*Figure 4E*). We observed no changes in IFN-γ levels in the absence of KVS-BI043 or normal T-cells (*Figure 4E*, gray boxed lanes). Most importantly, treatment of NCI-H716_A24 with KVS0001 in the presence of KVS-BI043 and T-cells led to significant killing of the target cancer cells, which was also not observed in the absence of T-cells or the absence of HLA-24 in the target cells

(*Figure 4F*). NCI-H2228 was assessed for killing but expressed too much TP53 at baseline (*Figure 4D*, left most lane) and thus displayed substantial killing even in the DMSO controls.

## Tumor growth is slowed in mice treated with KVS0001

Finally, we investigated whether KVS0001 could impact tumor growth in syngeneic models in which the native immune system might play a role (*Supplementary file 1*). For this purpose, we first used murine RENCA (renal cancer) and LLC (lung cancer) as they are known to have a relatively large number of out-of-frame indel mutations (*Supplementary file 5*). Although murine and human SMG1 are highly related (98% at the amino acid level), it was important to show that KVS0001 could actually inhibit NMD in murine cells. For this purpose, we tested eight genes in LLC, and four in RENCA, which contained out-of-frame indel mutations potentially targeted by NMD. We also assessed the expression of three genes without any mutations known to have their normal expression controlled by NMD (*Echols et al., 2020*). We found that six of the twelve truncating mutation-containing genes and five of the six expression controlled by NMD genes had significantly increased RNA following in vitro treatment with KVS0001 (*Figure 5A and B*).

We then implanted LLC and RENCA cancer cells in the mammary fat pad of C57BL/6N and BALB/c mice, respectively, and treated with KVS0001 or vehicle control (*Figure 5C*). The dose of KVS0001 was based on experiments showing that the maximum solubility limit was reached around 2–3 mg/ml, leading to a maximum single dose of 30 mg/kg per mouse per treatment. At this dose, the only toxicity noted was transient weight loss (*Figure 5—figure supplement 1*), and no other pathology was observed. Both tumor types experienced significant slowing of tumor growth (*Figure 5D*). However, when the same tumors were implanted in immunocompromised mice, there was no statistically significant difference in tumor growth between mice treated with KVS0001 or vehicle control (*Figure 5E*).

We evaluated a total of seven syngeneic mouse cancer cell lines. Four contained a relatively high number of truncating indel mutations, while the remaining three contained a moderate or low number. None of the three models with moderate or low numbers of truncating mutations responded to KVS0001 (EMT6, B16-F10, M3 Melanoma) (*Figure 5—figure supplement 2*). Of the four models with a higher number of truncating mutations, two (LLC and RENCA) responded in a statistically significant fashion while the other two failed to reach significance (CT26, MC38) (*Figure 5D* and *Figure 5—figure supplement 2*).

## Discussion

We report the design and execution of an HTS to identify small molecule inhibitors of the NMD pathway. The method employs a ratiometric output, allowing quantitative and controlled determination of RNA expression changes associated with NMD. Unlike previous NMD screens, this assay is directed at NMD as a process rather than as a way to identify compounds for treating specific mutations related to a disease state, such as cystic fibrosis or β-thalassemia (*Martin et al., 2014*; *Zhao et al., 2022*; *Salvatori et al., 2009*). The genetically modified isogenic cell lines employed here are uniquely suited for this approach, producing a high signal-to-noise ratio. Follow-up experiments confirmed a low false positive rate in the screen: four of the eight hits re-expressed transcripts from genes with truncating mutations in a dose-responsive manner when re-tested. Note that our assay allows for the detection of NMD inhibitors regardless of effects on protein translation by virtue of the comparison of wild-type to mutant transcripts for each queried gene. This is an important feature, as many previously reported inhibitors of NMD, such as emetine and anisomycin, rely on inhibition of protein translation and thus are unlikely to be optimal for specific restoration of NMD-targeted RNA and protein expression (*Siddiqui et al., 1973*; *Tang et al., 2012*; *Bongiorno et al., 2021*; *Bhuvanagiri et al., 2014*). During the review process, it was also noted that different NMD inhibitors caused re-expression of different knockout clones to different extents (see *Figure 1B* and *Figure 1—figure supplement 8* as an example). It is not clear to us why this occurred, but we speculate that it may be related to differences in the mechanism of action (i.e., protein synthesis inhibition-based NMD effects versus targeted NMD pathway inhibition as one example).

The optimal inhibitor of NMD found in our screen was LY3023414, and we suspected that this compound may inhibit the kinase SMG1 based on previously published kinase activity data with this small molecule (*Smith et al., 2016*). SMG1 acts as the gatekeeper of UPF1 activity through its

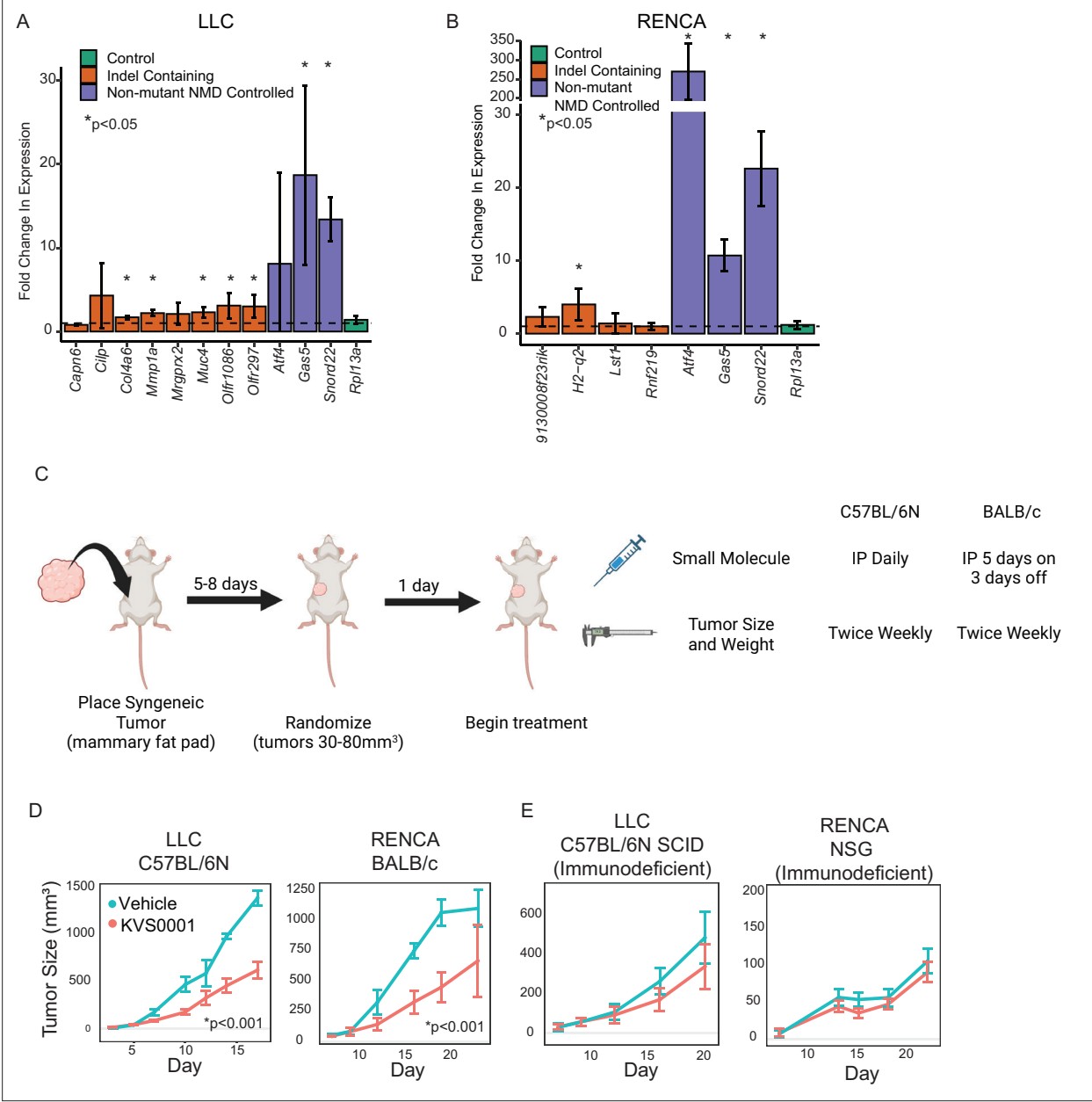

**Figure 5.** In vivo treatment of murine tumors with KVS0001 yield differential tumor growth compared with vehicle treatment. (**A**) Fold change in RNA transcript levels in LLC or (**B**) RENCA cells treated in vitro with 5 µM of nonsense-mediated decay (NMD) inhibitor KVS0001 or dimethyl sulfoxide (DMSO). Orange bars indicate genes with homozygous indel mutations potentially targeted by NMD. Purple bars show genes with no mutations but that are known to have their normal transcription levels controlled by NMD. Green bar is a control gene that should not change with treatment. The dotted line shows relative expression of DMSO treatment (equal to 1). * indicates significantly different from untreated by Student's T-test. (**C**) Treatment schedule for syngeneic tumor mouse experiments. (**D**) Average tumor size of LLC (left) and RENCA (right) syngeneic tumors in immune-competent mice ($n = 8$) treated with 30 mg/kg KVS0001 or vehicle control IP. Difference is statistically significant after day 10 based on one-way analysis of variance (ANOVA) with Dunnett's test $p < 0.001$ ($p < 0.05$ for day 23 RENCA data point) for both tumors tested. (**E**) Average tumor size of LLC (left) and RENCA (right) in immunodeficient mice ($n = 8$) treated with 30 mg/kg KVS0001 or vehicle control. Error bars show 95% confidence intervals in all plots.

The online version of this article includes the following figure supplement(s) for figure 5:

**Figure supplement 1.** Mouse weights for (**A**) C57BL/6N and (**B**) BALB/c tumor-bearing mice treated with 30 mg/kg KVS0001 or vehicle control intraperitoneal (IP) daily.

**Figure supplement 2.** Mouse tumor size as measured by calipers for mammary fat pad placed syngeneic tumor models treated daily with 30 mg/kg KVS0001 or vehicle IP.

phosphorylation at multiple sites (*Yamashita et al., 2001*). Unfortunately, issues likely related to off-target (pan-kinase) toxicity at the doses required for NMD inhibition prevented LY3023414 from being utilized in further in vivo work. While there are previously described SMG1 inhibitors, all have undesirable off-target or biophysical characteristics complicating their use as therapeutic agents (*Martin et al., 2014*; *Durand et al., 2007*; *Zhao et al., 2022*; *Gotham et al., 2016*; *Bongiorno et al., 2021*; *Becker et al., 2021*; *Cheruiyot et al., 2018*). For example, SMG1i-11 was previously identified as a potent and specific SMG1 inhibitor (*Gopalsamy et al., 2012*). However, in our hands, it was not soluble at concentrations required for in vivo work in a suitable vehicle for mouse administration. Another unrelated compound has shown successful in vivo re-expression of mutant RNA transcripts, but the effect is reported as less than a twofold change, compared to the three to fourfold demonstrated here by KVS0001, and little work has been done to develop the compound further in subsequent studies (*Keeling et al., 2013*).

Despite these chemistry-related challenges, our screen offered an unbiased assessment of the best targets for NMD pathway disruption, and thus we concluded that SMG1 was an ideal protein to inhibit for this purpose. In light of this, we attempted to improve the biophysical properties of SMG1i-11 through the development of KVS0001. As reported in this study, KVS0001 was a specific inhibitor of SMG1 that was soluble and could easily be administered to mice. Treatment of KVS0001 not only increased levels of transcripts and proteins from genes with truncating mutations, but also increased the levels of corresponding peptide bound–HLA complexes on the cell surfaces. We developed a new bispecific antibody, KVS-B043, which could recognize these pHLA complexes when the truncating mutations were present in the *TP53* gene and destroy cancer cells harboring such mutations. In syngeneic mouse models, KVS0001 slowed the growth of some tumors containing multiple truncating mutations, but only when the mice were immunocompetent. This dependence on an intact immune system supports the idea that the slowed growth of the treated tumors was due to recognition of neoantigens.

Despite these encouraging data, the potential utility of KVS0001 for therapeutic purposes remains speculative, for several reasons. First, we do not understand why KVS0001 (and other NMD inhibitors) increase the expression of many, but not all, genes with truncating mutations. We also do not systematically address the changes to splicing that are undoubtedly occurring when NMD is disrupted in either normal or tumor tissue. More research on the biochemistry of NMD will undoubtedly be required for such understanding, and perhaps the tools described here may facilitate that effort. Second, treatment with KVS0001 reduced the growth of some but not all syngeneic tumor types tested in mice. There appeared to be a correlation between the ability of KVS0001 to inhibit growth and the number of truncating mutations in the tumor. Likewise, recent evidence has been presented showing that cells deficient in known splicing factors are sensitive to subsequent NMD inhibition (*Cheruiyot, 2021*). This situation is reminiscent of that encountered with immune checkpoint inhibitors, in which there is a positive but imperfect correlation with loss of microsatellite instability-related proteins and tumor mutation burden (*Yarchoan et al., 2017*; *Ma et al., 2022*). Undoubtedly, the ability of the immune system to react to neoantigens is related to their quality as well as their quantity, with quality defined as the ability of the neoantigen to be bound to the host's particular MHC constitution and initiate an immune response (*Halima et al., 2022*; *Ribas and Wolchok, 2018*). Third, treatment with KVS0001 led to decreased tumor growth in immunocompetent mice, but did not lead to tumor regressions of the type mandated by RECIST criteria in human clinical trials (*Eisenhauer et al., 2009*; *Seymour et al., 2017*). Whether this failure to cure tumors in mice results from the contrived nature of the models used – injection of large numbers of rapidly growing cancer cells into animals with relatively little time to react prior to their demise – complicates the interpretation of many preclinical immunotherapeutic approaches.

Finally, KVS0001, though it was soluble and could be administered to animals for at least a month, was not entirely non-toxic, causing transient weight loss in some of the mice. Previous work has raised concerns that the importance of NMD in development and normal gene expression make it a difficult pathway to safely disrupt (*Mendell et al., 2004*; *Kurosaki et al., 2019*; *Tarpey et al., 2007*; *Nguyen et al., 2013*; *McIlwain et al., 2010*). The limited impact on body weight or observed physical activity of mice undergoing KVS0001 therapy at therapeutically relevant doses suggests that NMD inhibition may have acceptable toxicity and be tolerated in developed animals. Another example of this lies in the natural genetic variants of NMD components, which convey variability in the efficiency of NMD

between humans, and lends itself to supporting that knockdown of this pathway may be tolerable (*Linde et al., 2007*; *Nguyen et al., 2014*; *MacArthur et al., 2012*). Studies in mice also support this observation, as post-development knockdown of *UPF1* displayed minimal phenotype change (*Echols et al., 2020*). The broad use of NMD in normal development and growth, coupled with the observations here, suggests future work with inhibitors of this pathway should continue with a close eye for on-target off-tissue toxicity.

Though there is much work to be done, we hope that the tools, approaches, and compounds described here will facilitate that work. Should the administration of KVS0001 or a related compound prove non-toxic and well tolerated in humans, we end with a speculation about the possibility of KVS0001 being used as a preventative rather than as a therapeutic agent. Its potential use to prevent the onset of symptoms in pediatric syndromes caused by germline truncating mutations is obvious (*Lindeboom et al., 2019*; *Miller and Pearce, 2014*). Less obvious is the potential for it to prophy-lactically reduce cancer incidence in patients with hereditary non-polyposis colorectal cancer. These patients inherit heterozygous mutations of a mismatch repair gene, and they do not develop tumors until biallelic mutations of that mismatch repair gene are acquired in a rare stem cell during the second or third decade of life. Similar speculations can be made about the potential of KVS0001 to be used to prevent cancer initiation or progression in patients with other inherited mutations in repair genes, or in individuals exposed to high levels of exogenous mutagens.

# Materials and methods

## Key resources table

| Reagent type (species) or resource | Designation | Source or reference | Identifiers | Additional information |
|---|---|---|---|---|
| Antibody | Donkey anti goat IgG 680RD monoclonal | Licor | 926-68074, RRID:AB_10956736 | Used for western blot at 1:10 K |
| Antibody | Donkey anti mouse 680RD monoclonal | Licor | 926-68072, RRID:AB_10953628 | Used for western blot at 1:15 K |
| Antibody | Goat anti mouse HRP monoclonal | Jackson Immuno Research | 115-035-006, RRID:AB_2338500 | Used for western blot at 1:2500 |
| Antibody | Donkey anti rabbit 800CW monoclonal | Licor | 926-32213, RRID:AB_621848 | Used for western blot at 1:10 K |
| Antibody | Goat anti rabbit HRP monoclonal | Jackson Immuno Research | 111-035-006, RRID:AB_2337936 | Used for western blot at 1:2500 |
| Antibody | Mouse anti Beta-actin monoclonal | Cell Signaling Technology | 3700S | Used for western blot at 1:5000 |
| Antibody | Rabbit anti EXOC1 polyclonal | Abcam | ab251853 | Used for western blot at 0.4 µg/ml |
| Antibody | Mouse anti LMAN1 monoclonal | Thermo Fisher Scientific | CF502137 | Used for western blot at 1:200 |
| Antibody | Mouse anti OKT-3 monoclonal | Biolegend | 317347 | Used for T-cell culturing at 15 ng/ml |
| Antibody | Rabbit anti p21/WAF1 monoclonal | Cell Signaling Technology | 2947S, RRID:AB_823586 | Used for western blot at 1:1000 |
| Antibody | Mouse anti TP53 monoclonal | Cell Signaling Technology | 18032S, RRID:AB_2798793 | Used for western blot at 1:1000 |
| Antibody | Rabbit anti Phospho-(Ser/Thr) ATM/ATR polyclonal | Cell Signaling Technology | 2851S, RRID:AB_330318 | Used for western blot at 1:1000 |
| Antibody | Rabbit anti Sodium Potassium Pump (Na-K) polyclonal | Cell Signaling Technology | 3010S, RRID:AB_2060983 | Used for western blot at 1:500 |
| Antibody | Rabbit anti SPTAN1 (polyclonal) | Bethyl | A301-249A, RRID:AB_890655 | Used for western blot at 0.04 µg/ml |
| Antibody | Mouse anti SPTAN1 (N-Terminal) monoclonal | Abcam | ab11755, RRID:AB_298540 | Used for western blot at 1:1000 |
| Antibody | Rabbit anti SPTAN1 (C-Terminal) monoclonal | Abcam | ab75755, RRID:AB_1309947 | Used for western blot at 1:1000 |
| Antibody | Rabbit anti STAG2 monoclonal | Cell Signaling Technology | 5882S, RRID:AB_10834529 | Used for western blot at 1:1000 |
| Antibody | Goat anti UPF1 polyclonal | Abcam | ab10510, RRID:AB_297251 | Used for western blot at 1:2500 |

*Continued on next page*

*Continued*

| Reagent type (species) or resource | Designation | Source or reference | Identifiers | Additional information |
|---|---|---|---|---|
| Antibody | Mouse α-HLA-A24 monoclonal | MBL Life Science | Cat #K0208-A64, RRID:AB_1953030 | Used for flow cytometry at 10 µg/ml |
| Antibody | TP53-CD3 bispecific antibody | This study | N/A | See 'bispecific scFv construction' in methods, anti CD-3 and anti TP53 sequence. Used for co-culture at 12.5 pg/ml |
| Other | HLA-A24 Retrovirus | *Hsiue et al., 2021* | N/A | Retrovirus which introduces and HLA-A24 expression vector |
| Other | Luciferase lentivirus | OriGene | Cat #PS100071 | Lentivirus that introduces luciferase into cells |
| Chemical compound, drug | 1% penicillin–streptomycin | Thermo Fisher Scientific | Cat #15140122 | |
| Other | 4–15% Mini-PROTEAN TGX Precast Protein Gels | Bio-Rad | Cat #456-1086 | Used for western blotting |
| Chemical compound, drug | 5-Fluoruracil | Sigma-Aldrich | Cat #F6627 | |
| Other | AMPure beads | Beckman Coulter | Cat #A63880 | Used to purify DNA before sequencing |
| Chemical compound, drug | Cremaphor | Sigma-Aldrich | Cat #C5135 | |
| Chemical compound, drug | DharmaFECT1 transfection reagent | Horizon | Cat #T-2001-02 | |
| Chemical compound, drug | Dimethyl sulfoxide (DMSO) | Sigma-Aldrich | Cat #C6295 | |
| Chemical compound, drug | DMEM medium | Gibco | Cat #11995065 | |
| Chemical compound, drug | EMEM medium | ATCC | Cat #30-2003 | |
| Chemical compound, drug | Emetine | Sigma-Aldrich | Cat #7083-71-8 | |
| Chemical compound, drug | EPITHELIAL CELL MEDIUM-Complete Kit | Science Cell Research | Cat #4101 | |
| Chemical compound, drug | Fetal bovine serum (FBS) | HyClone | Cat #16777-006 | |
| Chemical compound, drug | Glycerol | Sigma-Aldrich | Cat #G5516 | |
| Chemical compound, drug | KVS0001 | This study | N/A | Novel small molecule targeting SMG1 kinase. See 'Resource availability' |
| Chemical compound, drug | LY3023414 | Selleckchem | Cat #S8322 | |
| Chemical compound, drug | Matrigel Phenol Red Free Standard Formulation | Corning | Cat #356237 | |
| Chemical compound, drug | MEGM Mammary Epithelial Cell Growth Medium BulletKit | Lonza | Cat #CC3150 | |
| Chemical compound, drug | Methylcellulose | Sigma-Aldrich | Cat #M6385 | |
| Chemical compound, drug | Phosphate-buffered saline (PBS) | Thermo Fisher | Cat #J60465.K2 | |
| Other | Phusion Flash High-Fidelity PCR Master Mix | Thermo Fisher | Cat #F548S | PCR reagent |
| Other | Pierce ECL Western Blotting Substrate | Thermo Fisher | Cat #32106 | Western blot reagent |
| Other | PRIMETIME Gene Expression master mix | Integrated DNA Technologies | Cat #1055770 | Real-time PCR reagent |
| Chemical compound, drug | Protease inhibitor | Millipore Sigma | Cat #4693159001 | |
| Other | QIAshredder | QIAGEN | Cat #79656 | Western blot reagent |
| Recombinant protein | Recombinant IL-2 protein | Prometheus Therapeutics and Diagnostics | Cat #aldesleukin | |
| Recombinant protein | Recombinant IL-7 protein | BioLegend | Cat #581908 | |

*Continued on next page*

*Continued*

| Reagent type (species) or resource | Designation | Source or reference | Identifiers | Additional information |
|---|---|---|---|---|
| Chemical compound, drug | RediJect D-Luciferin Ultra Bioluminescent Substrate | PerkinElmer | Cat #770505 | |
| Chemical compound, drug | RIPA Lysis and Extraction buffer | Thermo Fisher | Cat #89901 | Western blot and real-time PCR reagent |
| Chemical compound, drug | RNA Later | Invitrogen | Cat #AM7020 | Real-time PCR reagent |
| Chemical compound, drug | RPMI 1640 medium | Gibco | Cat #11875-119 | |
| Chemical compound, drug | Selleckchem Bioactive Compound library (**Supplementary file 2**) | Selleckchem | Cat #L1700 | |
| Chemical compound, drug | SMG1-specific inhibitor: 11j | Ascendex LLC | N/A | |
| Other | SsoAdvanced Universal SYBR Green Supermix | Bio-Rad | Cat #1725270 | Real-time PCR reagent |
| Chemical compound, drug | Trypsin | Gibco | Cat #25300054 | |
| Commercial assay or kit | AllPrep DNA/RNA Mini Kit | QIAGEN | Cat #80204 | |
| Commercial assay or kit | Agilent RNA ScreenTape | Agilent | Cat #5067-5576 | |
| Commercial assay or kit | Agilent RNA ScreenTape Sample Buffer | Agilent | Cat #5067-5577 | |
| Commercial assay or kit | Agilent RNA Ladder | Agilent | Cat #5067-5578 | |
| Commercial assay or kit | BCA Protein Assay Kit | Thermo Fisher | Cat #23227 | |
| Commercial assay or kit | Bio-Rad SingleShot Cell Lysis kit | Bio-Rad | Cat #1725080 | |
| Commercial assay or kit | ELISA | R&D Systems | Cat #SIF50C | |
| Commercial assay or kit | High-Capacity cDNA Reverse Transcription Kit | Advanced Biosystems | Cat #4368814 | |
| Commercial assay or kit | Illumina RNA library prep kit | Illumina | Cat #RS-122-2001 | |
| Commercial assay or kit | Kinase kinativ assay | ActivX Biosciences | N/A | |
| Commercial assay or kit | Luciferase Assay System | Promega | Cat #E1501 | |
| Commercial assay or kit | RNeasy | QIAGEN | Cat #74104 | |
| Other | Whole transcriptome RNA-seq: NCI-H358 | This study | | FASTQ files for RNA-seq. Available on Dryad |
| Other | Whole transcriptome RNA-seq: LS180 | This study | | FASTQ files for RNA-seq. Available on Dryad |
| Other | Whole transcriptome RNA-seq: Isogenic knockout cell lines | *Cook et al., 2022* | EGAD00001008559 | FASTQ files for RNA-seq. Available on EGA database |
| Cell Line (murine) | B16-F10 | ATCC | Cat #CRL-6475, RRID:CVCL_0159 | |
| Cell Line (Human) | HEK293T | ATCC | Cat #CRL-3216, RRID:CVCL_0063 | |
| Cell Line (murine) | Lewis lung carcinoma (LLC) | ATCC | Cat #CRL-1642, RRID:CVCL_4358 | |
| Cell Line (Human) | LS180 | ATCC | Cat #CL-187, RRID:CVCL_0397 | |
| Cell Line (Human) | MCF10a | ATCC | Cat #CRL-10317, RRID:CVCL_0598 | |
| Cell Line (Human) | NCI-H716 | ATCC | Cat #CCL-251, RRID:CVCL_1581 | |
| Cell Line (Human) | NCI-H2228 | ATCC | Cat #CRL-5935, RRID:CVCL_1543 | |
| Cell Line (Human) | NCI-H358 | ATCC | Cat #CRL-5807, RRID:CVCL_1559 | |
| Cell Line (murine) | Renca | ATCC | Cat #CRL-2947, RRID:CVCL_2174 | |

*Continued on next page*

*Continued*

| Reagent type (species) or resource | Designation | Source or reference | Identifiers | Additional information |
|---|---|---|---|---|
| Cell Line (Human) | RPE1 | ATCC | Cat #CRL-4000, RRID:CVCL_4388 | |
| Cell Line (Human) | RPE1 STAG2 2 | *Cook et al., 2022* | N/A | |
| Cell Line (Human) | RPE1 STAG2 5 | *Cook et al., 2022* | N/A | |
| Cell Line (Human) | RPE1 STAG2 6 | *Cook et al., 2022* | N/A | |
| Cell Line (Human) | RPE1 STAG2 8 | *Cook et al., 2022* | N/A | |
| Cell Line (Human) | RPE1 TP53 221 | *Cook et al., 2022* | N/A | |
| Cell Line (Human) | RPE1 TP53 223 | *Cook et al., 2022* | N/A | |
| Cell Line (Human) | RPE1 TP53 224 | *Cook et al., 2022* | N/A | |
| Cell Line (Human) | RPTec | ATCC | Cat #CRL-4031, RRID:CVCL_K278 | |
| Cell Line (Human) | RPTec STAG2 3 | *Cook et al., 2022* | N/A | |
| Cell Line (Human) | RPTec STAG2 943 | *Cook et al., 2022* | N/A | |
| Cell Line (Human) | RPTec STAG2 943A | *Cook et al., 2022* | N/A | |
| Cell Line (Human) | RPTec STAG2 944 | *Cook et al., 2022* | N/A | |
| Cell Line (Human) | RPTec STAG2 945 | *Cook et al., 2022* | N/A | |
| Cell Line (Human) | RPTec STAG2 946 | *Cook et al., 2022* | N/A | |
| Cell Line (Human) | RPTec STAG2 947 | *Cook et al., 2022* | N/A | |
| Cell Line (Human) | RPTec STAG2 951 | *Cook et al., 2022* | N/A | |
| Cell Line (Human) | RPTec STAG2 952 | *Cook et al., 2022* | N/A | |
| Cell Line (Human) | RPTec STAG2 953 | *Cook et al., 2022* | N/A | |
| Cell Line (Human) | RPTec TP53 544 | *Cook et al., 2022* | N/A | |
| Cell Line (Human) | RPTec TP53 588 | *Cook et al., 2022* | N/A | |
| Strain, strain background (*Mus musculus*, female) | BALB/cAnAHsd | Harlan Laboratories | | |
| Strain, strain background (*M. musculus*, female) | C57BL/6NCrl | Charles River GmbH | | |
| Strain, strain background (*M. musculus*, female) | BALB/cAnNCrl | Charles River GmbH | | |
| Strain, strain background (*M. musculus*, female) | Hsd:Athymic Nude-Foxn1$^{nu}$ | Harlan Laboratories | | |
| Strain, strain background (*M. musculus*, female) | NOD.Cg-Prkdcscid Il2rgtm1Wjl/SzJ (NSG) | Jackson Laboratories | | |
| Recombinant DNA reagent | for Primers see *Supplementary file 6* | This study | N/A | |
| Recombinant DNA reagent | HTS Screen sequencing primer: STAG2 Seq Forward: AATGATAC GGCGACCACCGA GATCTACACTCTTTCCCTACAC GACGCTCTTCCGATCT NNNNNNNNGAAT TTCTCTACAAAAAGCTCTTCA | This study | N/A | Used in high-throughput screen to amplify *STAG2* CRISPR mutation site |
| Recombinant DNA reagent | HTS Screen sequencing primer: STAG2 Seq Reverse: CAAGCAGAAGACGGCATACGAGAT NNNNNNNNNNTTCATCATTCCATCCTCCTC | This study | N/A | Used in high-throughput screen to amplify *STAG2* CRISPR mutation site |
| Recombinant DNA reagent | HTS Screen sequencing primer: TP53 Seq Forward: AATGATACGGCGAC CACCGAGATCTACACTCTT TCCCTACACGACGCTCTTC CGATCTNNNNNNNN GAAACTACTTCCTG AAAACAACGT | This study | N/A | Used in high-throughput screen to amplify *TP53* CRISPR mutation site |

*Continued on next page*

*Continued*

| Reagent type (species) or resource | Designation | Source or reference | Identifiers | Additional information |
|---|---|---|---|---|
| Recombinant DNA reagent | HTS Screen sequencing primer: TP53 Seq Reverse: CAAGCAGAAGACGGC ATACGAGATN NNNNNNNNNGCTTCATC TGGACCTGGGTC | This study | N/A | Used in high-throuput screen to amplify *TP53* CRISPR mutation site |
| Recombinant DNA reagent | ON-TARGETplus Human UPF1 siRNA Smartpool 10 nmol | Horizon Discovery | Cat #L-011763-00-0010 | |
| Recombinant DNA reagent | ON-TARGETplus Human mTOR siRNA Smartpool 10 nmol | Horizon Discovery | Cat #L-003008-00-0010 | |
| Recombinant DNA reagent | ON-TARGETplus Human ATM siRNA Smartpool 10 nmol | Horizon Discovery | Cat #L-003201-00-0010 | |
| Recombinant DNA reagent | ON-TARGETplus Human ATR siRNA Smartpool 10 nmol | Horizon Discovery | Cat #L-003202-00-0010 | |
| Recombinant DNA reagent | SMARTpool: ON-TARGETplus SMG1 siRNA | Horizon Discovery | Cat #L-005033-00-0020 | |
| Recombinant DNA reagent | SMARTpool: ON-TARGETplus PRKDC siRNA | Horizon Discovery | Cat #L-005030-00-0010 | |
| Recombinant DNA reagent | SMARTpool: ON-TARGETplus PIK3CA siRNA | Horizon Discovery | Cat #L-003018-00-0010 | |
| Recombinant DNA reagent | ON-TARGETplus Non-targeting Pool | Horizon Discovery | Cat #D-001810-10 | |
| Recombinant DNA reagent | Phage display library | GeneArt | N/A | |
| Software, algorithm | HISAT2 (version 2.0.5) | *Kim et al., 2019* | N/A | |
| Software, algorithm | StringTie (version 1.3.3) | *Pertea et al., 2016* | N/A | |
| Software, algorithm | Ballgown (version 2.6.0) | *Pertea et al., 2016* | N/A | |
| Software, algorithm | R (version 4.0.3) | *R Development Core Team, 2022* | N/A | |
| Software, algorithm | ggplot (version 3.4.1) | *Wickham, 2016* | N/A | |
| Software, algorithm | Pipeline for analyzing mutant and normal transcript abundance | MSSQL was used for initial data processing | N/A | |

## Resource availability

### Contact for reagent and resource sharing

Further information and requests for resources and reagents should be directed to and will be fulfilled by the lead contact, Nicolas Wyhs (wyhs@jhmi.edu). Isogenic knockout cell lines are available through The Genetic Resources Core Facility at Johns Hopkins School of Medicine (jhbiobank@jhmi.edu, Maryland, USA). KVS0001 generated in this study will be made available on request if available, but availability may be limited and we may require a payment and/or a completed materials transfer agreement if there is potential for commercial application.

## Experimental models and subject details

### Cell lines

NCI-H716, NCI-H2228, HEK293, NCI-H358, RPE1, MCF10a, RPTec, LLC, RENCA, B16-F10, and LS180 cells were purchased from The American Type Culture Collection (Virginia, USA). RPE1 STAG2 2, RPE1 STAG2 5, RPE1 STAG2 6, RPE1 STAG2 8, RPTec STAG2 3, RPTec STAG2 943, RPTec STAG2 943A, RPTec STAG2 944, RPTec STAG2 945, RPTec STAG2 946, RPTec STAG2 947, RPTec STAG2 951, RPTec STAG2 952, RPTec STAG2 953, RPE1 TP53 221, RPE1 TP53 223, RPE1 TP53 224, RPTec TP53 54, RPTec TP53 588, and RPTec TP53 544 isogenic knockout cell lines were generated and grown as previously described (*Cook et al., 2022*). NCI-H358, NCI-H2228, NCI-H716, RENCA, and RPE1 cells were grown in RPMI 1640 Medium (Gibco, California, USA, Cat #11875-119) supplemented with 10% fetal bovine serum (FBS) (HyClone, Utah, USA, Cat #16777-006). LS180 cells were grown in Eagle's Minimum Essential Medium (EMEM) (ATCC, Virginia, USA, Cat #30-2003) supplemented with 10% FBS. LLC, B16-F10, and HEK293 were grown in DMEM (Gibco, USA, Cat #11995065) supplemented with 10% FBS. RPTec cells were grown in EPITHELIAL CELL MEDIUM-Complete Kit (Science Cell

Research, California, USA, Cat #4101). MCF10a cells were grown in MEGM Mammary Epithelial Cell Growth Medium BulletKit (Lonza, USA, Cat #CC3150). In vitro, all cells were grown at 37°C with 5% $CO_2$. Mycoplasma testing was performed by The Genetic Resources Core Facility at Johns Hopkins School of Medicine (Maryland, USA).

## Quantitative real-time PCR

RNA was obtained from cells using a QIAGEN RNeasy Kit (QIAGEN, Maryland, USA, Cat #74104) per the manufacturer's instruction. Reverse transcription of RNA to cDNA was performed using a High-Capacity cDNA Reverse Transcription Kit (Applied Biosystems, USA, Cat #4368814) per the manufacturer's instructions. Unless otherwise specified, qPCR reactions were set up using SsoAdvanced Universal SYBR Green Supermix (Bio-Rad, USA, Cat #1725270) following the manufacturer's instructions and performed on a QuantStudio 3 (Applied Biosystems, USA) with the manufacturer's recommended plates and plate covers. PCR thermocycling conditions were as follows: 2 min at 50°C, 2 min at 95°C, 35 cycles of 10 s at 95°C, 10 s at 60°C, 30 s at 72°C. Finally, a melt curve was performed starting at 50°C and ending at 95°C with 5-s incubations for imaging. For TP53β analysis, qPCR reactions were set up using PRIMETIME Gene Expression master mix (Integrated DNA Technologies, USA, Cat #1055770) and a probe with FAM reporter and TAMRA quencher (Integrated DNA Technologies, USA) following the manufacturer's instructions and concentrations. PCR thermocycling conditions were as follows: 3 min at 50°C, 10 min at 95°C, 40 cycles of 10 s at 95°C, 30 s at 60°C. Primers used for all qPCR can be found in *Supplementary file 6*.

## Small molecule compounds

SMG1-specific inhibitor SMG1i-11 (11j) (*Gopalsamy et al., 2012*) and KVS0001 were synthesized by Ascendex LLC (Pennsylvania, USA). The synthesis scheme for KVS0001 is located below. Emetine was obtained from Sigma-Aldrich (Cat #7083-71-8). All other small molecule hits from the screen were purchased from Selleckchem (Texas, USA).

**Chemical structure 1.** Flowchart describing synthesis for KVS0001.

## NMD screen

We screened the Selleckchem Bioactive Compound library, a 2658 compound library (Selleckchem, Texas, USA, Cat #L1700). The library was distributed over 33 × 96-well tissue culture plates with each plate containing library compounds, two NMD positive controls (emetine, 12 µg/ml) and six NMD negative controls (DMSO). We mixed the three cell lines, RPE1 STAG2 2, RPE1 STAG2 8, and RPE1 TP53 221, in equal ratio and dosed with the compound library at 10 µM for 16 hr. Cell pools were harvested by washing plates 2× with phosphate-buffered saline (PBS) and frozen. RNA extraction and reverse transcriptase cDNA conversion were performed as described in *Targeted RNA-seq.* Quality control was performed by qPCR of one NMD-positive and one NMD-negative control well from each plate to confirm that increased expression of the positive control (emetine) wells was observed. All samples were prepared for sequencing by amplifying cDNA using primers containing both a plate (forward primer) and well (reverse primer) barcode which amplifies the knockout cell line-specific mutation for STAG2 and TP53 (see *Supplementary file 6*). Primers were obtained from Integrated DNA Technologies (Iowa, USA). PCR was performed using Phusion Flash High-Fidelity PCR Master Mix (Thermo Fisher, USA, Cat #F548S) for 1 min at 98°C, 29 cycles of 10 s at 98°C, 15 s at 64°C, 15 s at 72°C, and 5 min at 72°C. Samples were then well barcoded using a similar PCR setup for two to four cycles. Samples were then pooled, cleaned up with AMPure beads (Beckman Coulter, California, USA, Cat #A63880), and sequenced on an Illumina HiSeq2500 using the manufacturer's instructions

(150 cycle single read) for an average of 56,753 reads per well. The screen was scored by calculating the number of sequencing reads matching each mutant and reference transcript and calculating a mutant allele fraction (MAF) correcting for the number of cell lines and heterozygous mutations. NMD-positive and -negative control data were pooled, respectively, and averaged across all plates to determine the hit threshold of MAF >0.46. This value, 5 standard deviations above the mean for the DMSO controls, was chosen to ensure no false positives in the DMSO controls were observed. A compound was considered a hit if the MAF for all three cell lines was greater than this value. All results from the screen are reported as MAF.

## Targeted RNA-seq

RNA extraction of in vitro cells was performed using Bio-Rad SingleShot Cell Lysis kit (Bio-Rad, California, USA, Cat #1725080) scaled down to 50 µl reactions per well of 96-well plate per the manufacturer's instructions. Tissue culture cells were lysed directly on the tissue culture plate. For RNA extraction of in vivo studies, tissues were harvested and placed in RNA Later (Invitrogen, Maryland, USA, Cat # AM7020) and stored at −80°C until RNA extraction. RNA extraction of tissues was performed using a QIAGEN RNeasy Kit (QIAGEN, Maryland, USA, Cat #74104) per the manufacturer's instruction with tissue homogenization in 600 µl of RNA lysis buffer (RLT buffer) via dounce homogenizer followed by QIAshredder (QIAGEN, Maryland, USA, Cat #79656). All RNA quality was assessed by Agilent Tapestation 2200 (Agilent, California, USA, Cat #G2964AA) and the Agilent RNA ScreenTape (Agilent, California, USA, Cat #5067-5576) with Agilent RNA ScreenTape Sample Buffer and Ladder (Agilent, California, USA, Cat #5067-5577, Cat #5067-5578) per the manufacturer's instruction. Reverse transcription to cDNA was performed using High-Capacity cDNA Reverse Transcription Kit (Applied Biosystems, USA, Cat #4368814) per the manufacturer's instructions. cDNA was amplified using cDNA-specific primers with at least one primer (forward or reverse) covering an exon–exon boundary (see *Supplementary file 6* for primer sequences). Primers were obtained from Integrated DNA Technologies (Iowa, USA). PCR was performed using Phusion Flash High-Fidelity PCR Master Mix (Thermo Fisher, USA, Cat #F548S) for 1 min at 98°C, 30 cycles of 10 s at 98°C, 15 s at 64°C, 15 s at 72°C, and 5 min at 72°C. Sequencing libraries were prepped from samples by addition of well barcodes using the same method described above for an additional two to six cycles. Libraries were pooled, cleaned up with AMPure beads (Beckman Coulter, California, USA, Cat #A63880) and sequenced on an Illumina Miseq using the manufacturer's instructions (150 cycle single read).

## MAF analysis

MAF was determined by processing fastq files using HISAT2 (version 2.0.5) and aligning to a pseudo reference genome consisting of only the mutant or wild-type amplicon sequences for targeted regions. The MAF was determined by taking a ratio of the number of mutant transcripts to the total number of transcripts from the region in question. Initial data processing was performed in MSSQL and Excel.

## Whole transcriptome RNA-seq

LS180 or NCI-H358 cells were run in biological duplicate, treated with DMSO or 5 µM LY3023414. For RNA extraction, cells were pelleted, frozen in liquid nitrogen, and stored at −80°C until RNA extraction. RNA extraction was performed using a QIAGEN AllPrep DNA/RNA Mini Kit (QIAGEN, Maryland, USA, Cat #80204) per the manufacturer's instruction with cell homogenization and lysis in RLT buffer with a QIAshredder (QIAGEN, Maryland, USA, Cat #79656). RNA quality control using Agilent Tapestation 2200 (Agilent, California, USA, Cat #G2964AA) and the Agilent RNA ScreenTape (Agilent, California, USA, Cat #5067-5576) with Agilent RNA ScreenTape Sample Buffer and Ladder (Agilent, California, USA, Cat #5067-5577, Cat #5067-5578) per the manufacturer's instruction. Library prep using Illumina RNA library prep kit (Illumina, California, USA, Cat #RS-122–2001) and sequenced on an Illumina HiSeq 4000 150 cycle paired-end using the manufacturer's instructions.

## RNA-seq analysis

Sequencing reads aligned to Hg38 using HISAT2 (version 2.0.5), RNA alignment metrics using CollectRnaSeqMetrics (Picard, version 2.20.2). Exon skipping was determined using IGV Viewer Sashimi Plots (*Robinson et al., 2011*). The average number of bases sequenced per sample and percent

aligned in LS180 was 5.14e9 bases (range 5.07e9–5.20e9) and 77.8% (range 76.1–78.7%) and for NCI-H358 was 5.58e9 bases (5.19e9–5.87e9) and 79.4% (range 77.2–82.7%). MAF was determined using VarScan 2 (**Koboldt et al., 2012**) by generating the ratio of Read 1 (mutant) to read2 (wild-type) transcripts. Mutations were only considered if they were heterozygous, and contained at least five reads at the somatic mutation or indel site in all four samples being compared (biological duplicates of treated and untreated).

## Compound response curves

We performed a 6-point dose–response curve by treating cell pools for 14 hr with single compounds. Cell pools consisted of three isogenic knockout cell lines grouped by parental cell line with readout via targeted sequencing RNA (see *Targeted RNA-seq* for details). The effect of the compound was determined by calculating the MAF by comparing the abundance of the expected mutation and compared to the wild-type within each isogenic pool (see *MAF analysis* for details).

## Immunohistochemistry

Immunohistochemistry was performed on cell lines as previously described (**Hsiue et al., 2021**).

## Western blots

Cells were lysed using radioimmunoprecipitation assay buffer (RIPA buffer) (Thermo Fisher, USA, Cat #89901) containing 1× protease inhibitor (Thermo Fisher, USA, Cat #4693159001) on ice for 30 min. Samples were then centrifuged at max speed for 3 min at 4°C in a QIA shredder (QIAGEN, Maryland, USA, Cat #79654) before being transferred to a new sample collection tube. Protein was quantified using a bicinchoninic acid (BCA) assay (Thermo Fisher, USA, Cat #23227) per the manufacture's instructions. Gels were run by loading 50 µg of total protein per sample into 15-well polyacrylamide gels (Bio-Rad, California, USA, Cat #456-1086) and run for 30 min at 200 V. Gels were then transferred using the manufacturer's instructions (based on size) to nitrocellulose membrane using a Bio-Rad turbo transfer apparatus (Bio-Rad, USA, #170-4270). Membranes were blocked for 1 hr with 3% milk-TBS-Tween before being incubated overnight in primary antibody (concentration dependent on antibody). Primary antibodies and concentrations can be found in the Key resources table. Membranes were washed four times for 5 min each with TBS-Tween. Secondary antibody was applied at 1:2500 using either α-rabbit (Jackson ImmunoResearch, Pennsylvania, USA, Cat #111-035-006) or α-mouse (Jackson ImmunoResearch, Pennsylvania, USA, Cat #115-035-006). Membranes were imaged using Pierce ECL Western Blotting Substrate (Thermo Fisher, USA, Cat #32106) following the manufacturer's instructions on a Bio-Rad Chemidoc (Bio-Rad, California, USA). Phospho-UPF1 westerns were performed as detailed above with the following exceptions: secondary antibodies were used at 1:10,000 either α-rabbit (Licor Biosciences, USA, Cat #926-32213) or α-goat (Licor Biosciences, USA, Cat #926-68074). Membranes were imaged using an Odyssey CLx (Licor Biosciences, USA) following the manufacturer's instructions.

## siRNA

We performed knockdown of kinase proteins using siRNA. LS180 and NCI-H358 cells were plated in 96-well plates (Costar, USA, Cat #3595) at 5000 cells per well. Cells were transfected with Dharma-FECT1 transfection reagent (Horizon, USA, Cat #T-2001-02) and either 50 nM control or kinase-specific pooled siRNA. siRNA used in this experiment can be found in the Key resources table. Knockdown efficacy was confirmed using qPCR (see *Quantitative real-time PCR* for details). NMD target gene mutation transcript levels were determined using NGS and MAF (see *Targeted RNA-seq* for details).

## Bispecific scFv construction

We utilized a bispecific antibody against CD-3 and TP53 wild-type peptide 'TYSPALNKMF' (residues 125–134) presented in a HLA-A24 MHC-I molecule scFv. This bispecific antibody was identified by panning a phage display library. The scFv-bearing phage library was constructed similarly as described in detail previously with some modifications (**Skora et al., 2015**). Briefly, oligonucleotides were synthesized by GeneArt (Thermo Fisher, USA) using trinucleotide mutagenesis (TRIM) technology to diversify complementarity-determining region (CDR)-L2, CDR-L3, CDR-H1, CDR-H2, and CDR-H3. A

FLAG (DYKDDDDK) epitope tag was placed immediately downstream of the scFv, which was followed in frame by the full-length M13 pIII coat protein sequence. The total number unique clones obtained was determined to be $3.6 \times 10^{10}$. Panning details can be found in the reference section (*Hsiue et al., 2021*; *Skora et al., 2015*).

## Kinase target assay

Kinase kiNativ experiments were performed by ActivX Biosciences (San Diego, USA) (*Patricelli et al., 2011*).

## Quantitative presentation of HLA-bound neoantigens via HPLC–mass spectrometry

Identification and quantitation of HLA-presented neoantigens were performed as previously described by Complete Omics Inc (*Wang et al., 2019*). Briefly, cells were treated in vitro for 24 hr with either DMSO or 5 µM KVS0001. Cells were cross-linked and immunoprecipitated with pan-HLA antibodies to obtain cell surface MHC-presented peptides. Mass spectrometry was performed in the presence of heavy labeled peptide to serve as an internal (loading) control to quantify the presence of the presented neoantigen. The presented peptides were identified in a preliminary MS screen of 187 candidate peptides predicted by using the union of NetMHC and Predictor of Immunogenic Epitopes (PRIME) predictions (*Reynisson et al., 2020*; *Schmidt et al., 2021*).

## CD3-TP53scFv bispecific co-culture

Co-culture of bispecific antibody was performed using volunteer human donor T-cells. T-cell enrichment and activation were performed as previously described (*Hwang et al., 2021*). Briefly, peripheral blood mononuclear cells (PBMCs) are incubated with OKT3 antibody (Biolegend) for 3 days. T-cells were then expanded in RPMI 1640 with 10% FBS and 1% penicillin–streptomycin, recombinant IL-2 (Proleukin, Prometheus Laboratories) and IL-7 (BioLegend) for at least 15 days before use. NCI-H716 cells were labeled with HLA-A24 using retrovirus as previously described (*Hsiue et al., 2021*). Briefly, the MSCV retroviral expression system (Clontech, USA, Cat #634401) was used to overexpress HLA-A*24-T2A-GFP in target cells. Expression was confirmed by flow cytometry. NCI-H2228 cells were not labeled as they express this specific HLA endogenously. For NCI-H716_A24 and NCI-H2228, 40,000 cells were plated in 96-well plates, and co-incubated with 40,000 activated T-cells, and 12.5 pg/ml of TP53-CD3 bispecific antibody. Cells were then dosed with 1.25 µM of KVS0001, SMG1-specific inhibitor or 200 mg/ml of 5-fluoruracil (Sigma-Aldrich, USA, Cat #F6627) for 24 hr. Readout of IFN-γ was performed by ELISA following the manufacturer's instructions (R&D Systems, USA, Cat #SIF50C). Cell killing was assayed by luciferase levels following the manufacturer's instructions (Promega, USA, Cat #E1501) and read out on a Synergy H1 Microplate reader (BioTek, USA). NCI-H716_A24 cells were labeled with luciferase via lentiviral transduction following the manufacturer's recommendations (OriGene, USA, Cat #PS100071).

## Animal protocols

Animal research was approved and overseen by Johns Hopkins University Institutional Animal Care and Use Committee (IACUC) approved research protocol M018M79. Mice are housed in individually ventilated caging (Allentown, New Jersey, USA) at a maximum 5 animals per cage. Cages are changed every 14 days. Enrichment is provided through paper bedding, paper hut, and some food placed in the bottom of the cage. Facility is maintained between 70 and 72°F on a 12-hr light–dark cycle. Mice standard diet is ad lib Teklad Global 18% Protein Extruded Rodent Diet, autoclaved (Envigo, Huntingdon, UK) and acidified water via sipper tube.

## In vivo tumor models

Cells for tumor inoculation in each mouse were grown to 70–80% confluency in vitro. Cells were harvested with trypsin (Gibco, California, USA, Cat #25300054) and suspended in either PBS for LS180 cells or 50% PBS, 50% Matrigel Phenol Red Free Standard Formulation (Corning, New York, USA, Cat #356237) for NCI-H358 cells. For both cell lines, 1e6 cells were placed subcutaneously and grown to approximately 200 mm³. Animals were randomized into treatment groups prior to treatment by

tumor size. All tumor volumes were measured via caliper twice weekly. Mice with tumors <150 mm$^3$ at 11 days post-inoculation were excluded from experiments.

## Human xenograft experiments

Six- to eight-week-old *Mus musculus* Hsd:Athymic Nude-Foxn1$^{nu}$ mice (referred to as nude mice) were purchased from Harlan Laboratories (Indiana, USA). Only female mice were used as gender was not considered to be a significant confounder in the experiment. LS180 cells were inoculated at $1.0 \times 10^6$ cells per mouse in the left mouse flank or NCI-H358 cells were inoculated at $7.5 \times 10^5$ cells per mouse in the right flank. For single-dose experiments, mice were orally dosed via gavage at 0, 40, or 60 mg/kg of LY3023414 (Selleckchem, Texas, USA, Cat #S8322) in 1% methylcellulose (Sigma-Aldrich, Missouri, USA, Cat #M6385). KVS0001 was dosed at 30 mg/ml intraperitoneal (IP) in 0.5% DMSO (Sigma-Aldrich, USA, Cat #C6295), 10% cremaphor (Sigma-Aldrich, USA, Cat #C5135), and 2% glycerol (Sigma-Aldrich, USA, Cat #G5516). Mice were given physical exams prior to euthanasia at designated endpoints according to the approved research protocol. Tumors, spleen, blood, and lungs were harvested post-euthanasia for further analysis. Tumors were harvested at indicated times and stored in RNA Later (Invitrogen, Maryland, USA, Cat # AM7020) at −80 for further analysis.

## Syngeneic mouse tumor experiments

Six- to eight-week-old immune compromised *M. musculus* BALB/cAnNCrl (referred to as BALB/c) and C57BL/6NCrl (referred to as C57BL/6N) mice were purchased from Charles River GmbH (Germany). Only female mice were used as gender was not considered to be a significant confounder in the experiment. Murine cancer cell lines were placed in the left mammary fat pad of female mice at the following densities: LLC $1.0 \times 10^6$, RENCA $1.0 \times 10^6$, and B16-F10 $0.2 \times 10^6$. Mice were randomized around day 7 post-implantation when tumor sizes had reached approximately 30–40 mm$^3$. Mice were dosed with 30 mg/kg KVS0001 or vehicle control via IP injection alternating the right and left side daily for 28 days (*Figure 5C*). Mice were given physical exams before euthanasia at designated endpoints according to the approved research protocol. Syngeneic mouse tumor experiments were repeated offsite by a private contract research organization (CRO), Reaction Biology, blinded to the drug they were providing. Results shown in this manuscript are results as obtained by this CRO.

## In vivo TP53 bispecific experiments

For the NCI-H716 in vivo bispecific experiments $2.5 \times 10^6$ NCI-H716_A24 expressing cells were placed orthotopically by IP injection into NOD.Cg-Prkdcscid Il2rgtm1Wjl/SzJ (referred to as NSG) mice (Jackson Laboratories, USA). On day 2 mice were randomized to ensure a balanced tumor burden in all groups. Luminescence was measured by injecting mice with 150 µl of RediJect D-Luciferin Ultra Bioluminescent Substrate (PerkinElmer, USA, Cat #770505), and anesthetized using isoflurane in an induction chamber for 5 min. Readouts and analysis were performed on an IVIS Spectrum imaging system and Living Image software (PerkinElmer, USA).

## Data reporting

No statistical methods were used to predetermine the sample size. Statistical analysis was performed using R and Excel. All animal experiments were randomized being sure to alter the first cage dosed and location of cages with a row of the rack. Randomization was performed by animal tumor size, unless otherwise indicated. The investigators responsible for weight and tumor measurements were blinded to the allocation, treatment, and outcome assessment of experiments. The investigators processing mouse tissue processing were blinded to the allocation, treatment, and outcome assessment of experiments. The investigator harvesting the tissues was aware of the allocation, treatment, and outcome assessment of experiments.

## Statistical testing

Chi-squared testing was performed using counts of cancer cell line mutations predicted to undergo NMD and RNA recovered and prop.test() in R. Mann–Whitney testing performed with wilcox.test() in R. Quantiles calculated utilizing $z$ scores with quantile() in R. Student $t$-test and one-way ANOVA with Dunnett's test and Student's $t$-test were performed in R and Excel.

## Acknowledgements

The authors thank Dr. Qing Wang from Complete Omics Inc for his help and advice with the neoantigen presentation by mass spectrometry studies. We would also like to thank Dr. Xiao Chen from Ascendex LLC. for helpful suggestions during medicinal chemistry experiments. Biophysical predictions for small molecules were done using Molinspiration Cheminformatics free web services, https://www.molinspiration.com, Slovensky Grob, Slovakia, accessed 9/2023. This work was supported by Oncology Core CA 06973 (BV, KWK, NP); the Virginia and D K Ludwig Fund for Cancer Research (BV, KWK, NP, CB); the Sol Goldman Sequencing Facility at Johns Hopkins (BV); NIH Cancer Center Support Grant P30 CA006973. SP was supported by NCI Grant K08CA270403, the Leukemia and Lymphoma Society Translation Research Program Award, the American Society of Hematology Scholar Award, and the Swim Across America Translational Cancer Research Award.

## Additional information

### Competing interests

Surojit Sur: Consultant for CAGE Pharma. Provisional patent applications on the work described in this paper have been filed by Johns Hopkins University under accession 63/451,738. Joshua D Cohen: Owns equity in Haystack Oncology. Provisional patent applications on the work described in this paper have been filed by Johns Hopkins University under accession 63/451,738. Bert Vogelstein: Founder of Thrive Earlier Detection, an Exact Sciences Company. Hold equity in Exact Sciences. Founder of or consultants to and own equity in ManaT Bio., Haystack Oncology, Neophore, CAGE Pharma and Personal Genome Diagnostics. Consultant to and holds equity in Catalio Capital Management. Provisional patent applications on the work described in this paper have been filed by Johns Hopkins University under accession 63/451,738. Nickolas Papadopoulos: Founder of Thrive Earlier Detection, an Exact Sciences Company. Consultant to Thrive Earlier Detection. Hold equity in Exact Sciences. Founders of or consultants to and own equity in ManaT Bio., Haystack Oncology, Neophore, CAGE Pharma and Personal Genome Diagnostics. Consultant to Vidium. Provisional patent applications on the work described in this paper have been filed by Johns Hopkins University under accession 63/451,738. Chetan Bettegowda: Consultant to Depuy-Synthes, Bionaut Labs, Haystack Oncology, Privo Technologies and Galectin Therapeutics. Co-founder of OrisDx and Belay Diagnostics. Shibin Zhou: Hold equity in Exact Sciences. Founders of or consultants to and own equity in ManaT Bio., Haystack Oncology, Neophore, CAGE Pharma and Personal Genome Diagnostics. Has a research agreement with BioMed Valley Discoveries, Inc. Kenneth W Kinzler: Founders of Thrive Earlier Detection, an Exact Sciences Company. Consultants to Thrive Earlier Detection. Hold equity in Exact Sciences. Founders of or consultants to and own equity in ManaT Bio., Haystack Oncology, Neophore, CAGE Pharma and Personal Genome Diagnostics. Provisional patent applications on the work described in this paper have been filed by Johns Hopkins University under accession 63/451,738. The other authors declare that no competing interests exist.

### Funding

| Funder | Grant reference number | Author |
|---|---|---|
| Howard Hughes Medical Institute | | Bert Vogelstein |
| Ludwig Family Foundation | | Bert Vogelstein<br>Kenneth W Kinzler |
| Oncology Core CA | 06973 | Bert Vogelstein<br>Nickolas Papadopoulos<br>Kenneth W Kinzler |
| Sol Goldman Charitable Trust | | Bert Vogelstein |

| Funder | Grant reference number | Author |
| --- | --- | --- |
| NCI Cancer Center Support Grant | P30 CA006973 | Bert Vogelstein<br>Nickolas Papadopoulos<br>Chetan Bettegowda<br>Kathy Gabrielson<br>Shibin Zhou<br>Kenneth W Kinzler |
| National Cancer Institute | K08CA270403 | Suman Paul |
| Leukemia and Lymphoma Society | Translation Research Program Award | Suman Paul |
| American Society of Hematology | Scholar Award | Suman Paul |
| Swim Across America | Translational Cancer Research Award | Suman Paul |

The funders had no role in study design, data collection, and interpretation, or the decision to submit the work for publication.

## Author contributions

Ashley L Cook, Conceptualization, Resources, Data curation, Formal analysis, Validation, Investigation, Visualization, Methodology, Writing – original draft, Writing – review and editing; Surojit Sur, Conceptualization, Formal analysis, Supervision, Methodology; Laura Dobbyn, Blair Ptak, Bum Seok Lee, Data curation, Investigation; Evangeline Watson, Resources, Data curation; Joshua D Cohen, Data curation, Software, Formal analysis; Suman Paul, Investigation, Methodology; Emily Hsiue, Resources; Maria Popoli, Data curation, Formal analysis, Investigation; Bert Vogelstein, Conceptualization, Formal analysis, Supervision, Funding acquisition, Visualization, Methodology, Writing – original draft, Project administration, Writing – review and editing; Nickolas Papadopoulos, Chetan Bettegowda, Supervision, Funding acquisition, Project administration; Kathy Gabrielson, Resources, Formal analysis, Investigation, Methodology; Shibin Zhou, Conceptualization, Formal analysis, Supervision, Funding acquisition, Project administration; Kenneth W Kinzler, Conceptualization, Formal analysis, Supervision, Funding acquisition, Investigation, Methodology, Writing – original draft, Project administration, Writing – review and editing; Nicolas Wyhs, Conceptualization, Resources, Data curation, Formal analysis, Supervision, Validation, Investigation, Visualization, Methodology, Writing – original draft, Project administration, Writing – review and editing

## Author ORCIDs

Ashley L Cook https://orcid.org/0000-0001-6789-5732
Surojit Sur https://orcid.org/0000-0003-4536-7343
Chetan Bettegowda https://orcid.org/0000-0001-9991-7123
Nicolas Wyhs https://orcid.org/0000-0003-4252-4470

## Ethics

Animal research was approved and overseen by Johns Hopkins University Institutional Animal Care and Use Committee (IACUC) approved research protocol M018M79. Mice are housed in individually ventilated caging (Allentown, New Jersey, USA) at a maximum 5 animals per cage. Cages are changed every 14 days. Enrichment is provided through paper bedding, paper hut, and some food placed in the bottom of the cage. Facility is maintained between 70 and 72°F on a 12-hr light–dark cycle. Mice standard diet is ad lib Teklad Global 18% Protein Extruded Rodent Diet, autoclaved (Envigo, Huntingdon, UK) and acidified water via sipper tube.

Reviewer #1 (Public Review): https://doi.org/10.7554/eLife.95952.3.sa1
Reviewer #2 (Public Review): https://doi.org/10.7554/eLife.95952.3.sa2
Author response https://doi.org/10.7554/eLife.95952.3.sa3

## Additional files

### Supplementary files

Supplementary file 1. Summary of cell lines used in this manuscript and their origin.

Supplementary file 2. Selleck Chem Library compounds used for the nonsense-mediated decay (NMD) high-throughput screen (HTS screen).

Supplementary file 3. Structures of novel nonsense-mediated decay (NMD; SMG1) inhibitors designed for this study.

Supplementary file 4. List of kiNativ assay results for all kinase proteins tested at three concentrations of KVS0001.

Supplementary file 5. Mutations found in LLC and RENCA murine tumors.

Supplementary file 6. List of primers used in this study with sequences.

MDAR checklist

### Data availability

Whole transcriptome RNA-seq raw and processed data have been deposited at Dryad. Full western blot images have been deposited at Mendeley. Any additional information required to reanalyze the data reported in this paper is available from the lead contact upon request.

The following datasets were generated:

| Author(s) | Year | Dataset title | Dataset URL | Database and Identifier |
|---|---|---|---|---|
| Wyhs N | 2024 | NMD Paper Full Westerns | https://doi.org/10.17632/pyzxxvjphk.4 | Mendeley Data, 10.17632/pyzxxvjphk.4 |
| Wyhs N | 2025 | RNA-seq data for LS180 and NCI-H358 Cancer Cell Lines | https://doi.org/10.5061/dryad.69p8cz9dj | Dryad Digital Repository, 10.5061/dryad.69p8cz9dj |

The following previously published dataset was used:

| Author(s) | Year | Dataset title | Dataset URL | Database and Identifier |
|---|---|---|---|---|
| Cook AL, Wyhs N, Sur S, Ptak B, Popoli M, Dobbyn L, Papadopoulos T, Bettegowda C, Papadopoulos N, Vogelstein B, Zhou S, Kinzler KW | 2022 | TSGKO_RNASeq | https://ega-archive.org/datasets/EGAD00001008559 | European Genome-Phenome Archive, EGAD00001008559 |

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
