## [Editor Report · eLife assessment]

Here, the authors developed a cell-based screening assay for the identification of small molecule inhibitors of nonsense-mediated decay (NMD), and used it to validate KVS0001, a new small molecule SMG1 kinase inhibitor derived from the existing inhibitor SMG1i-11, showing it inhibits NMD in cultured cells leading to expression of neoantigens from NMD-targeted genes and slows tumor growth of cancer cell lines possessing a significant number of out-of-frame indel mutations. The conclusions are supported by **convincing** evidence, and the significance of this work consists in the development of a new and very promising NMD inhibitor drug that acts as an inhibitor of the SMG1 NMD kinase and is effective in animal tumor studies. This is an **important** advance for the field, as previous NMD inhibitors were not specific, lacked efficacy, or were very toxic and hence not suitable for animal applications.

---

## [Referee Report · Reviewer #1 (Public Review)]

Summary:

This work identified new NMD inhibitors and tested them for cancer treatment, based on the hypothesis that inhibiting NMD could lead to the production of cancer neoantigens from the stabilized mutant mRNAs, thereby enhancing the immune system's ability to recognize and kill cancer cells. Key points of the study include:

• Development of an RNA-seq based method for NMD analysis using mixed isogenic cells that express WT or mutant transcripts of STAG2 and TP53 with engineered truncation mutations.

• Application of this method for a drug screen and identified several potential NMD inhibitors.

• Demonstration that one of the identified compounds, LY3023414, inhibits NMD by targeting the SMG1 protein kinase in the NMD pathway in cultured cells and mouse xenografts.

• Due to the in vivo toxicity observed for LY3023414, the authors developed 11 new SMG1 inhibitors (KVS0001-KVS0011) based on the structures of the known SMG1 inhibitor SMG1i-11 and the SMG1 protein itself.

• Among these, KVS0001 stood out for its high potency, excellent bioavailability and low toxicity in mice. Treatment with KVS0001 caused NMD inhibition and increased presentation of neoantigens on MHC-I molecules, resulting in the clearance of cancer cells in vitro by co-cultured T cells and cancer xenografts in mice by the immune system.

These findings support the strategy of targeting the NMD pathway for cancer treatment and provide new research tools and potential lead compounds for further exploration.

Strengths:

The RNA-seq based NMD analysis, using isogenic cell lines with specific NMD-inducing mutations, represents a novel approach for the high-throughput identification of potential NMD modulators or genetic regulators. The effectiveness of this method is exemplified by the identification of a new activity of AKT1/mTOR inhibitor LY3023414 in inhibiting NMD.

The properties of KVS0001 described in the manuscript as a novel SMG1 inhibitor suggest its potential as a lead compound for further testing the NMD-targeting strategies in cancer treatment. Additionally, this compound may serve as a useful research tool.

The results of the in vitro cell killing assay and in vivo xenograft experiments in both immuno-proficient and immune-deficient mice indicate that inhibiting NMD could be a viable therapeutic strategy for certain cancers.

Weaknesses:

The authors did not address the potential effects of NMD/SMG1 inhibitors on RNA splicing. Given that the transcripts of many RNA-binding proteins are natural targets of NMD, inhibiting NMD could significantly alter splicing patterns. This, in turn, might influence the outcomes of the RNA-seq-based method for NMD analysis and result interpretation.

While the RNA-seq based approach offers several advantages for analyzing NMD, the effects of NMD/SMG1 inhibitors observed through this method should be confirmed using established NMD reporters. This step is crucial to rule out the possibility that mutations in STAG2 or TP53 affect NMD in cells, as well as to address potential clonal variations between different engineered cell lines.

The results from the SMG1/UPF1 knockdown and SMG1i-11 experiments presented in Figure 3 correlate with the effects seen for LY3023414, but they do not conclusively establish SMG1 as the direct target of LY3023414 in NMD inhibition. An epistatic analysis with LY3023414 and SMG1-knockdown is needed.

Comment on the revised version:

Although KVS0001 exhibits promising properties as an SMG1 inhibitor for cancer treatment, it remains unclear if it is superior to existing SMG1 inhibitors, as no direct comparisons have been made.

---

## [Referee Report · Reviewer #2 (Public Review)]

Summary:

Several publications during the past years provided evidence that NMD protects tumor cells from being recognized by the immune system by suppressing the display of neoantigens, and hence NMD inhibition is emerging as a promising anti-cancer approach. However, the lack of an efficacious and specific small molecule NMD inhibitor with suitable pharmacological properties is currently a major bottleneck in the development of therapies that rely on NMD inhibition. In this manuscript, the authors describe their screen for identifying NMD inhibitors, which is based on isogenic cell lines that either express wild-type or NMD-sensitive transcript isoforms of p53 and STAG2. Using this setup, they screened a library of 2658 FDA-approved or late-phase clinical trial drugs and had 8 hits. Among them they further characterized LY3023414, showing that it inhibits NMD in cultured cells and in a mouse xenograft model, where it, however, was very toxic. Because LY3023414 was originally developed as a PI3K inhibitor, the authors claim that it inhibits NMD by inhibiting SMG1. While this is most likely true, the authors do not provide experimental evidence for this claim. Instead, they use this statement to switch their attention to another previously developed SMG1 inhibitor (SMG1i-11), of which they design and test several derivatives. Of these derivatives, KVS0001 showed the best pharmacological behavior. It upregulated NMD-sensitive transcripts in cultured cells and the xenograft mouse model, and two predicted neoantigens could indeed be detected by mass spectrometry when the respective cells were treated with KVS0001. A bispecific antibody targeting T cells to a specific antigen-HLA complex led to increased IFN-gamma release and killing of cancer cells expressing this antigen-HLA complex when they were treated with KVS0001. Finally, the authors show that renal (RENCA) or lung cancer cells (LLC) were significantly inhibited in tumor growth in immunocompetent mice treated with KVS0001. Overall, this establishes KVS0001 as a novel and promising ant-cancer drug that by inhibiting SMG1 (and therewith NMD) increases the neoantigen production in the cancer cells and reveals them to the body's immune system as "foreign".

Strengths:

The novelty and significance of this work consist in the development of a novel and - judging from the presented data - very promising NMD inhibiting drug that is suitable for applications in animals. This is an important advance for the field, as previous NMD inhibitors were not specific, lacked efficacy, or were very toxic and hence not suitable for animal application. It will be still a long way with many challenges ahead towards an efficacious NMD inhibitor that is safe for use in humans, but KVS0001 appears to be a molecule that bears promise for follow-up studies. In addition, while the idea of inhibiting NMD to trigger neoantigen production in cancer cells and so reveal them to the immune system has been around for quite some time, this work provides ample and compelling support for the feasibility of this approach, at least for tumors with a high mutational burden.

Main weaknesses:

There is a disconnect between the screen and the KVS0001 compound, that they describe and test in the second part of the manuscript since KVS0001 is a derivative of the SMG1 inhibitors developed by Gopalsamy et al. in 2012 and not of the lead compound identified in the screen (LY3023414). Because of high toxicity in the mouse xenograft experiments, the authors did not follow up LY3023414 but instead switched to the published SMG1i-11 drug of Gopalsamy and colleagues, a molecule that is widely used among NMD researchers for NMD inhibition in cultured cells. Therefore, in my view, the description of the screen is obsolete, and the paper could just start with the optimization of the pharmacological properties of SMG1i-11 and the characterization of KVS0001. Even though the screen is based on an elegant setup and was executed successfully, it was ultimately a failure as it didn't reveal a useful lead compound that could be further optimized.

Additional points:

- Compared to SMG1i-11, KVS0001 seems less potent in inhibiting SMG1 (higher IC50). It would therefore be important to also compare the specificity of both drugs for SMG1 over other kinases at the actually applied concentrations (1 uM for SMG1i-11, 5 uM for KVS0001). The Kinativ Assay (Fig. S13) was performed with 100 nM KVS0001, which is 50-fold less than the concentration used for functional assays and hence not really meaningful. In addition, more information on the pharmacokinetic properties and toxicology of KVS0001 would allow a better judgment of the potential of this molecule as a future therapeutic agent.

- On many figures, the concentrations of the used drugs are missing. Please ensure that for every experiment that includes drugs, the drug concentration is indicated.

- Do the authors have an explanation for why LY3023414 has a much stronger effect on the p53 than on the STAG2 nonsense allele (Fig. 1B, S8), whereas emetine upregulates the STAG2 nonsense alleles more than the p53 nonsense allele (Fig. S5). I find this curious, but the authors do not comment on it.

- While it is a strength of the study that the NMD inhibitors were validated on many different truncation mutations in different cell lines, it would help readers if a table or graphic illustration was included that gives an overview of all mutant alleles tested in this study (which gene, type of mutation, in which cell type). In the current version, this information is scattered throughout the manuscript.

- Lines 194 and 302: That SMG1i-11 was highly insoluble in the hands of the authors is surprising. It is unclear why they used variant 11j, since variant 11e of this inhibitor is widely used among NMD researchers and readily dissolves in DMSO.

- Line 296: The authors claim that they were able to show that LY3023414 inhibited the SMG1 kinase, which is not true. To show this, they would have for example to show that LY3023414 prevents SMG1-mediated UPF1 phosphorylation, as they did for KVS0001 and SMG1i-11 in Fig. 3F. Unless the authors provide this data, the statement should be deleted or modified.

Comments on the revised version:

- The authors have satisfactorily addressed all my "Additional points" listed above.

- With the new publishing model of Life, the authors ultimately decide on whether or not to follow reviewers suggestions, and in this case, the authors decided (against my suggestion) to leave the screening part in the manuscript, although it did not result in a useful lead compound. They argue it helped them define in an unbiased way SMG1 as the ideal target for NMD disruption. I would counterargue that this has been known in the field for quite a while.

- One last suggestion I have to the authors would be to modify the statement in the abstract "This led to the design of a novel SMG1 inhibitor", because what they call "novel" is, in reality, a chemical improvement of the pharmacological properties of a previously reported SMG1 inhibitor (Gopalsamy et al., 2012).

---

## [Author Response]

The following is the authors’ response to the original reviews.

**Public Reviews:**

**Reviewer #1 (Public Review):**
Summary:This work identified new NMD inhibitors and tested them for cancer treatment, based on the hypothesis that inhibiting NMD could lead to the production of cancer neoantigens from the stabilized mutant mRNAs, thereby enhancing the immune system's ability to recognize and kill cancer cells. Key points of the study include:• Development of an RNA-seq based method for NMD analysis using mixed isogenic cells that express WT or mutant transcripts of STAG2 and TP53 with engineered truncation mutations.• Application of this method for a drug screen and identified several potential NMD inhibitors.• Demonstration that one of the identified compounds, LY3023414, inhibits NMD by targeting the SMG1 protein kinase in the NMD pathway in cultured cells and mouse xenografts.• Due to the in vivo toxicity observed for LY3023414, the authors developed 11 new SMG1 inhibitors (KVS0001-KVS0011) based on the structures of the known SMG1 inhibitor SMG1i-11 and the SMG1 protein itself.• Among these, KVS0001 stood out for its high potency, excellent bioavailability, and low toxicity in mice. Treatment with KVS0001 caused NMD inhibition and increased presentation of neoantigens on MHC-I molecules, resulting in the clearance of cancer cells in vitro by co-cultured T cells and cancer xenografts in mice by the immune system.These findings support the strategy of targeting the NMD pathway for cancer treatment and provide new research tools and potential lead compounds for further exploration.Strengths:The RNA-seq-based NMD analysis, using isogenic cell lines with specific NMD-inducing mutations, represents a novel approach for the high-throughput identification of potential NMD modulators or genetic regulators. The effectiveness of this method is exemplified by the identification of a new activity of AKT1/mTOR inhibitor LY3023414 in inhibiting NMD.The properties of KVS0001 described in the manuscript as a novel SMG1 inhibitor suggest its potential as a lead compound for further testing the NMD-targeting strategies in cancer treatment. Additionally, this compound may serve as a useful research tool.The results of the in vitro cell killing assay and in vivo xenograft experiments in both immuno-proficient and immune-deficient mice indicate that inhibiting NMD could be a viable therapeutic strategy for certain cancers.Weaknesses:The authors did not address the potential effects of NMD/SMG1 inhibitors on RNA splicing. Given that the transcripts of many RNA-binding proteins are natural targets of NMD, inhibiting NMD could significantly alter splicing patterns. This, in turn, might influence the outcomes of the RNA-seq-based method for NMD analysis and result interpretation.

This is a very important comment that highlights an important aspect of NMD and potential exciting downstream studies. We did not systematically assess RNA splicing in our work as we are not sure if inhibition of NMD would induce cancer specific splicing that would allow for tumor targeting. It is well established that NMD can impact splicing, including modulating cryptic exon expression, but finding and assessing antigenicity of targetable tumor specific antigens constitutes a study in and of its own. Our own data in figure 4C-F supports this, as a point mutation near a splice site in TP53 strongly induced NMD which was subsequently stopped by KVS0001 treatment. Doing a systematic review of this effect we feel is outside the scope of this manuscript. We’ve incorporated a comment into our discussion highlighting this deficiency, but certainly find the idea of mining RNA-splicing changes an exciting next endeavor.

While the RNA-seq-based approach offers several advantages for analyzing NMD, the effects of NMD/SMG1 inhibitors observed through this method should be confirmed using established NMD reporters. This step is crucial to rule out the possibility that mutations in STAG2 or TP53 affect NMD in cells, as well as to address potential clonal variations between different engineered cell lines.

This is possible, but we want to highlight that all hits from the screen were confirmed in a separate cell line with different clones. While this will not rule out effects to NMD due to STAG2 and TP53 knockdown, the final lead compound was also tested on different endogenous transcripts in both indel and normal transcripts controlled by NMD (i.e., ATF4) in multiple species (human and mouse). Importantly, many of these assays employed the non-mutated transcripts from heterozygous mutant cells to ensure that cis-acting NMD was being measured and to control for any trans-acting splicing or other unanticipated biochemical effects.

The results from the SMG1/UPF1 knockdown and SMG1i-11 experiments presented in Figure 3 correlate with the effects seen for LY3023414, but they do not conclusively establish SMG1 as the direct target of LY3023414 in NMD inhibition. An epistatic analysis with LY3023414 and SMG1-knockdown is needed.

This is a great comment, and is supported by the recent push to confirm drug targets by chemical probes or knockout followed by loss of further effect due to the application of the drug in question. We attempted to knockout SMG1 in multiple cells lines used in this study, including RPE1, MCF10A, NCI-H358 and LS180, and were unable to obtain clones that have biallelic out of frame indels. We were able to obtain multiple clones with in frame indels. Based on our results and those in the publicly available database DepMap we suspect this gene is likely essential, making a simple knockout unfeasible. While this uncertainty is important to keep in mind, we feel it does not detract from the reporting of a novel NMD screen that is mechanistically agnostic and of a novel in vivo active NMD inhibitor.

**Reviewer #2 (Public Review):**
Summary:Several publications during the past years provided evidence that NMD protects tumor cells from being recognized by the immune system by suppressing the display of neoantigens, and hence NMD inhibition is emerging as a promising anti-cancer approach. However, the lack of an efficacious and specific small-molecule NMD inhibitor with suitable pharmacological properties is currently a major bottleneck in the development of therapies that rely on NMD inhibition. In this manuscript, the authors describe their screen for identifying NMD inhibitors, which is based on isogenic cell lines that either express wild-type or NMD-sensitive transcript isoforms of p53 and STAG2. Using this setup, they screened a library of 2658 FDA-approved or late-phase clinical trial drugs and had 8 hits. Among them they further characterized LY3023414, showing that it inhibits NMD in cultured cells and in a mouse xenograft model, where it, however, was very toxic. Because LY3023414 was originally developed as a PI3K inhibitor, the authors claim that it inhibits NMD by inhibiting SMG1. While this is most likely true, the authors do not provide experimental evidence for this claim. Instead, they use this statement to switch their attention to another previously developed SMG1 inhibitor (SMG1i-11), of which they design and test several derivatives. Of these derivatives, KVS0001 showed the best pharmacological behavior. It upregulated NMD-sensitive transcripts in cultured cells and the xenograft mouse model and two predicted neoantigens could indeed be detected by mass spectrometry when the respective cells were treated with KVS0001. A bispecific antibody targeting T cells to a specific antigen-HLA complex led to increased IFN-gamma release and killing of cancer cells expressing this antigen-HLA complex when they were treated with KVS0001. Finally, the authors show that renal (RENCA) or lung cancer cells (LLC) were significantly inhibited in tumor growth in immunocompetent mice treated with KVS0001. Overall, this establishes KVS0001 as a novel and promising ant-cancer drug that by inhibiting SMG1 (and therewith NMD) increases the neoantigen production in the cancer cells and reveals them to the body's immune system as "foreign".Strengths:The novelty and significance of this work consists in the development of a novel and - judging from the presented data - very promising NMD inhibiting drug that is suitable for applications in animals. This is an important advance for the field, as previous NMD inhibitors were not specific, lacked efficacy, or were very toxic and hence not suitable for animal application. It will be still a long way with many challenges ahead towards an efficacious NMD inhibitor that is safe for use in humans, but KVS0001 appears to be a molecule that bears promise for follow-up studies. In addition, while the idea of inhibiting NMD to trigger neoantigen production in cancer cells and so reveal them to the immune system has been around for quite some time, this work provides ample and compelling support for the feasibility of this approach, at least for tumors with a high mutational burden.Main weaknesses:There is a disconnect between the screen and the KVS0001 compound, that they describe and test in the second part of the manuscript since KVS0001 is a derivative of the SMG1 inhibitors developed by Gopalsamy et al. in 2012 and not of the lead compound identified in the screen (LY3023414). Because of high toxicity in the mouse xenograft experiments, the authors did not follow up LY3023414 but instead switched to the published SMG1i-11 drug of Gopalsamy and colleagues, a molecule that is widely used among NMD researchers for NMD inhibition in cultured cells. Therefore, in my view, the description of the screen is obsolete, and the paper could just start with the optimization of the pharmacological properties of SMG1i-11 and the characterization of KVS0001. Even though the screen is based on an elegant setup and was executed successfully, it was ultimately a failure as it didn't reveal a useful lead compound that could be further optimized.

This is a helpful observation from an outside perspective. From our point of view, we were only alerted to the targeting SMG1 due to the previously reported off-target effects of LY3023414 on SMG and lack of plausible explanation for PIK3CA inhibition to efficiently inhibit NMD. We do feel that the screen is worth including for two reasons. First, it offers an unbiased approach for querying the entire NMD pathway for vulnerabilities useful to target. The library chosen was quite small, so the screen itself could be useful to others with larger libraries to test. Second, it did help identify SMG1 as the ideal target for NMD disruption. While targeting SMG1 is not novel, we felt it highlighted why we chose to develop KVS0001. To address this reviewer’s comment, we’ve included a couple sentences in the results and discussion strengthening the point that the screen provided an unbiased approach to finding the best target in the pathway to disrupt NMD and elaborating on the transition from LY3023414 and the screen to development of KVS0001.

Additional points:- Compared to SMG1i-11, KVS0001 seems less potent in inhibiting SMG1 (higher IC50). It would therefore be important to also compare the specificity of both drugs for SMG1 over other kinases at the applied concentrations (1 uM for SMG1i-11, 5 uM for KVS0001). The Kinativ Assay (Fig. S13) was performed with 100 nM KVS0001, which is 50-fold less than the concentration used for functional assays and hence not really meaningful. In addition, more information on the pharmacokinetic properties and toxicology of KVS0001 would allow a better judgment of the potential of this molecule as a future therapeutic agent.

We agree that the Kinativ assay may have poorly represented the activity of KVS0001 at the bioactive concentration. We have now added 1uM Kinativ data, the highest concentration we were able to run to figure S13.

- On many figures, the concentrations of the used drugs are missing. Please ensure that for every experiment that includes drugs, the drug concentration is indicated.

We apologize for this oversight and have added all drug concentrations on the appropriate plots.

- Do the authors have an explanation for why LY3023414 has a much stronger effect on the p53 than on the STAG2 nonsense allele (Figure 1B, S8), whereas emetine upregulates the STAG2 nonsense alleles more than the p53 nonsense allele (Figure S5). I find this curious, but the authors do not comment on it.

This is an interesting observation. The short answer is we’re not sure. The speculative answer is that it is related to the distinctly different mechanisms of actions of the two inhibitors (see comments from reviewing editor below).

- While it is a strength of the study that the NMD inhibitors were validated on many different truncation mutations in different cell lines, it would help readers if a table or graphic illustration was included that gives an overview of all mutant alleles tested in this study (which gene, type of mutation, in which cell type). In the current version, this information is scattered throughout the manuscript.

This is an excellent suggestion. We’ve included a new table S1 which incorporates the details of each cell line and the genes used in each for this study.

- Lines 194 and 302: That SMG1i-11 was highly insoluble in the hands of the authors is surprising. It is unclear why they used variant 11j, since variant 11e of this inhibitor is widely used among NMD researchers and readily dissolves in DMSO.

As this referee notes SMG1i-11 is soluble in DMSO in our hands as well, which enabled us to use it for our in vitro work. Unfortunately, the concentrations of DMSO required to dissolve the compound to suitable concentrations for in vivo work were too high to safely use in mice with our animal protocols. We also attempted to use ethanol, which also did dissolve SMG1i-11, but led to a significant amount of toxicity in both the drug and vehicle control arms.

- Line 296: The authors claim that they were able to show that LY3023414 inhibited the SMG1 kinase, which is not true. To show this, they would have for example to show that LY3023414 prevents SMG1-mediated UPF1 phosphorylation, as they did for KVS0001 and SMG1i-11 in Fig. 3F. Unless the authors provide this data, the statement should be deleted or modified.

We’ve modified this statement as requested by the referee, now saying we suspected SMG1 was the target based on previously published work.

**Recommendations for the authors:**

**Reviewing Editor (Recommendations For The Authors):**
Your paper has been assessed by two reviewers with expertise in the NMD field. They both find the identification and characterization of a new potent and selective inhibitor of the SMG1 NMD kinase with in vivo activity to represent a significant advance in the field, and one that could ultimately be of value as the basis for a novel cancer therapy. However, as you will see both reviewers have concerns about whether the SMG1 inhibitor screen you developed belongs in the paper because it was not used to identify the KVS0001 inhibitor, which instead was generated based on a previously published set of SMG1 inhibitors, and because the NMD inhibitor that did emerge from your screen, LY3023414, was not shown to be a direct inhibitor of SMG1 kinase activity. While it is an elegant screen, during the revision of the paper you could consider streamlining the manuscript by emphasizing how the screening assay was used to validate KVS0001, and bolstering the characterization of the new KVS0001 NMD inhibitor by conducting the proposed additional experiments.Each of the reviewers raises additional points that should be addressed in a revised version.The reviewing editor has two additional points:(1) While emetine inhibits NMD, it is not really a direct NMD inhibitor, as implied, but rather a potent protein synthesis elongation inhibitor that acts by binding to the E-site of the 40S ribosomal subunit, and is therefore, like anisomycin, another protein synthesis inhibitor, working indirectly to inhibit NMD. This should be acknowledged in the section where emetine is first used as an "NMD inhibitor".

This has been included in the indicated section at the referee’s request.

(2) To establish that the observed phenotypic effects of KVS0001 are due to on-target inhibition of SMG1, the authors could generate and express an SMG1 point mutant that is resistant to KVS0001 inhibition, which could be based on the SMG1 catalytic domain structure that the authors used originally to design KVS001. Inhibitor-resistant kinase mutants are the gold standard for demonstrating that the biological consequences of a novel protein kinase inhibitor are due to on-target effects. Admittedly, because SMG1 is such a huge protein, this may be technically challenging and is likely beyond the scope of the present paper.

-We agree with the reviewing editor on all accounts: this would be an ideal experiment to run, but also that it is beyond the scope of the present paper. As indicated in our discussion above with reviewer 1, SMG1 knockout was not possible in our hands, and we suspect it may be due to the gene being essential. Creating an inhibitor resistant mutant could overcome this issue and create an ideal model to test the target for KVS0001. Unfortunately finding such a mutant would likely require significant amounts of trial and error to create a resistant mutant that did not lose SMG1 function. And SMG1 is huge, creating technical issues for experimenting. Due to the anticipated amount of work for such a study we believe this would be better accomplished in future studies.

**Reviewer #1 (Recommendations For The Authors):**
(1) The authors did not mention a new SMG1 inhibitor and its effects described in Cheruiyot et al, Cancer Res 2019 (PMID: 34215620).

A comment regarding this discovery and its implications for our work was added to the discussion.

(2) There is an inconsistency between the manuscript text and methods sections regarding the time of drug treatment (16 hours vs 14 hours) in the HTS screen.

This has been double checked in our notebook and fixed to reflect 16hrs as the correct incubation time. Thank you for identifying that clerical oversight.

**Reviewer #2 (Recommendations For The Authors):**
(1) Line 61: The references to NMD reviews are very old (refs 20 and 21). I suggest citing more recent, up-to-date reviews instead.

Two additional references, one from 2016 and another from 2023, have been added to increase support for this statement in the introduction.

(2) Figure S1: Shouldn't the caption of the right panel (TP32 data) say "clone 221" rather than "clone 22"?

This has been fixed.

(3) Figure S18: Please indicate on the y-axis that you are displaying RPKM for p53.

This has been fixed.

(4) Figures 4D and S19: Please indicate concentrations used for all drugs.

This has been fixed.